# Regulating the proximity effect of heterocycle-containing AIEgens

Jianyu Zhang [1,6], Yujie Tu[1,6], Hanchen Shen[1], Jacky W. Y. Lam [1], Jianwei Sun [1], Haoke Zhang [2,3] ✉ & Ben Zhong Tang [1,4,5] ✉

Proximity effect, which refers to the low-lying $(n,\pi^*)$ and $(\pi,\pi^*)$ states with close energy levels, usually plays a negative role in the luminescent behaviors of heterocyclic luminogens. However, no systematic study attempts to reveal and manipulate proximity effect on luminescent properties. Here, we report a series of methylquinoxaline derivatives with different electron-donating groups, which show different photophysical properties and aggregation-induced emission behaviors. Experimental results and theoretical calculation reveal the gradually changed energy levels and different coupling effects of the closely related $(n,\pi^*)$ and $(\pi,\pi^*)$ states, which intrinsically regulate proximity effect and aggregation-induced emission behaviors of these luminogens. With the intrinsic nature of heterocycle-containing compounds, they are utilized for sensors and information encryption with dynamic responses to acid/base stimuli. This work reveals both positive and negative impacts of proximity effect in heterocyclic aggregation-induced emission systems and provides a perspective to develop functional and responsive luminogens with aggregation-induced emission properties.

Materials science drives the advancement of the world and matters, especially for the versatile types of organic luminescent materials (OLMs), which have been developed and widely applied to optoelectronic devices, energy conversion, biological imaging, chemical sensors, etc[1–6]. Based on the well-established theory of through-bond conjugation (TBC), molecular photophysics has been developed and utilized to design and construct OLMs by covalently connecting conjugated units and introducing electron donors and acceptors[7–11]. However, most of the traditional OLMs possess planar and rigid skeletons and suffer from aggregation-caused quenching (ACQ) effect in the aggregate and solid states because of the strong π-π stacking, hindering their practical applications as solid devices[12–14].

Fortunately, aggregation-induced emission (AIE), an opposite effect to ACQ, was coined, which referred to the phenomenon that luminogens were strongly emissive in the aggregate state but nonemissive in their dilute solutions[15–17]. Unlike planar ACQ molecules, luminogens with AIE effect (AIEgens) show twisted or easily distorted conformation, and the restriction of intramolecular motion (RIM) is proven as the mechanism for AIE effect[18–21]. The development and application of AIEgens have been a hotspot for OLMs during the past two decades, and many AIEgens are constructed using the star fragments of tetraphenylethene or triphenylamine as the fundamental units[22–24]. Many TBC-based strategies have been applied to manipulate the luminescent properties of these π-conjugated systems. For example,

[1]Department of Chemistry, Hong Kong Branch of Chinese National Engineering Research Center for Tissue Restoration and Reconstruction, and Guangdong-Hong Kong-Macau Joint Laboratory of Optoelectronic and Magnetic Functional Materials, The Hong Kong University of Science and Technology, Clear Water Bay, Kowloon, Hong Kong 999077, China. [2]MOE Key Laboratory of Macromolecular Synthesis and Functionalization, Department of Polymer Science and Engineering, Zhejiang University, Hangzhou 310058, China. [3]Zhejiang-Israel Joint Laboratory of Self-Assembling Functional Materials, ZJU-Hangzhou Global Scientific and Technological Innovation Center, Zhejiang University, Hangzhou 311215, China. [4]School of Science and Engineering, Shenzhen Institute of Aggregate Science and Technology, The Chinese University of Hong Kong, Shenzhen, Guangdong 518172, China. [5]AIE Institute, Guangzhou Development District, Huangpu, Guangzhou 510530, China. [6]These authors contributed equally: Jianyu Zhang, Yujie Tu. ✉e-mail: zhanghaoke@zju.edu.cn; tangbenz@cuhk.edu.cn

the introduction of donor–acceptor (D–A) units endows AIEgens with twisted intramolecular charge transfer (TICT) properties to realize polarity- and moisture-responsive luminescence[25–28]. Hence, the design strategy and mechanistic understanding of the π-conjugated AIEgens are well-established and widely utilized for advanced applications.

Due to the complex electronic structures of heteroatom-participated compounds, many heteroatom-containing AIEgens from natural products or synthetic laboratories have attracted much attention in fabricating OLMs[29–33]. Meanwhile, the participation of heteroatoms and lone pairs brings some unique structures and properties that are absent in pure hydrocarbonic AIEgens, such as hydrogen bonds, excited-state intramolecular proton transfer, and room-temperature phosphorescence[34–37]. Negligently but importantly, some compounds also show unique properties closely related to the $(n,\pi^*)$ transition introduced by heteroatoms, such as the proximity effect (PE). PE refers to the phenomenon that the energy level of the low-lying $(n,\pi^*)$ state is close to the lowest $(\pi,\pi^*)$ state in some nitrogen-heterocyclic and aromatic carbonyl compounds, which was first proposed by Edward C. Lim et al. in the 1970s[38–41]. The vibronic coupling of these two states usually leads to efficient internal conversion and nonradiative decay, resulting in the quenching of luminogens[42–45]. Hence, PE is traditionally regarded as a negative factor in constructing OLMs. Several studies have reported the solvent effect, temperature-dependent fluorescence, and substituent effect related to PE[46–48]. However, no systematic study attempts to reveal and manipulate the PE of heterocycle-containing AIEgens. It is believed that the combination of PE and RIM effect would build new theories to replenish AIE mechanisms and provide strategies to develop efficient AIEgens with unique features.

Herein, a series of heterocycle-containing AIEgens with different electron-donating units were developed and synthesized to investigate the regulation of PE (Supplementary Figs. 1, 2). All compounds were satisfactorily characterized, and their photophysical properties were systematically investigated (Supplementary Figs. 3–12). They all showed typical AIE properties but different trends of fluorescence variation toward water addition during the aggregation process, which was modulated by the response of PE and TICT to solvent environments. The theoretical analysis clearly illustrated the electronic-structure change and different coupling effects of the low-lying $(n,\pi^*)$ and $(\pi,\pi^*)$ states on their photophysical properties, suggesting both positive and negative impacts of PE. In addition, with the intrinsic nature of heteroatoms and PE, these compounds exhibited sensitive fluorescence responses to acid/base stimuli, which were rationally utilized for sensor and information encryption. This work realizes the regulation of PE in heterocyclic AIE systems and provides a perspective to develop functional and responsive AIEgens.

## Results

### Photophysical properties

The photophysical properties of the heterocycle-containing core, 3-methylquinoxalin-2(1H)-one (MQ), were first studied (Fig. 1). It showed a maximum absorption peak ($\lambda_{abs}$) at 337 nm with a shoulder at 355 nm in different solvents without a noticeable solvent effect (Supplementary Fig. 13a). The photoluminescence (PL) property was measured in THF/water mixtures by stepwise adding water. As shown in Fig. 1b, MQ was almost nonemissive in pure THF solution but gradually became emissive along with the increased water fractions ($f_w$). At $f_w = 60\%$, the PL intensity ($I$) was 9-fold higher than that in pure THF solution ($I_0$) (Fig. 1c). Meanwhile, the maximum emission wavelength ($\lambda_{em}$) was redshifted from 393 nm to 410 nm. When $f_w \geq 70\%$, nanoparticles with a size around 100 nm were detected using the dynamic light scattering technique (Supplementary Fig. 14), suggesting the formation of aggregates. At $f_w = 90\%$, the PL intensity was further enhanced to 13-fold higher than $I_0$, and the $\lambda_{em}$ showed a slight redshift. The absolute

quantum yield ($\Phi$) of MQ in the crystalline state (2.1%) was much higher than the pure solution (0.1%) due to multiple hydrogen-bonding interactions and RIM effect, which was verified by its single-crystal structure (Supplementary Figs. 15, 16 and Supplementary Table 1). The above results clearly supported the AIE-active nature of MQ.

Since MQ is inherently an electron acceptor, the phenyl group as a weak electron donor is incorporated into MQ to construct (E)-3-styrylquinoxalin-2(1H)-one (SQ) with a D–A structure. Accordingly, its $\lambda_{abs}$ was redshifted to 390 nm and displayed a weak solvent effect in different solvents (Supplementary Fig. 13b). The PL spectra of SQ also showed gradually enhanced intensity with the increased $f_w$, but its PL intensity enhancement at $f_w = 90\%$ ($I_{90}/I_0 = 4$) was much smaller than MQ ($I_{90}/I_0 = 13$) (Fig. 1d, e). Besides, the $\lambda_{em}$ exhibited an unobvious redshift (from 454 nm to 458 nm) when $f_w \leq 70\%$ and rapidly shifted to 475 nm at $f_w = 90\%$. The first stage-redshift was similar to that of MQ, but the second-stage dramatic redshift showed quite different properties with MQ, which might be ascribed to the intermolecular π–π interactions in the closely packed aggregates, supported by its crystal packing and solid-state PL spectra with the same shoulder peak (Supplementary Figs. 15, 17). It is noteworthy that the redshifted $\lambda_{em}$ was also observed in the crystalline MQ, although the redshift was absent in the aggregate state at $f_w = 90\%$. The above results suggested that the addition of a weak donor (phenyl ring) not only changed its physical properties (solubility, packing structure, etc.) but also significantly altered the electronic structures and photophysical properties reflected by the redshifted $\lambda_{em}$ and weaker PL intensity enhancement.

Inspired by the above conclusion, the methoxy group and triphenylamine group with a stronger electron-donating ability were introduced to construct (E)-3-(4-methoxystyryl)quinoxalin-2(1H)-one (MeOSQ) and (E)-3-(4-(diphenylamino)styryl)quinoxalin-2(1H)-one (DPASQ), respectively (Fig. 2a). The $\lambda_{abs}$ of MeOSQ and DPASQ further shifted bathochromically to 400 nm and 440 nm, respectively (Supplementary Fig. 13). Typically, the absorption spectra of DPASQ showed an obvious solvent effect without fine structures, indicating its strong charge transfer (CT) feature. Their PL spectra were investigated and compared in THF/water mixtures. Similar to SQ, MeOSQ displayed enhanced PL intensity and redshifted $\lambda_{em}$ with the increased $f_w$, which could also be divided into two stages by $f_w = 60\%$ (Fig. 2b, c). These two stages should be manipulated by the polarity of the solvent environment and aggregation effects, respectively (Supplementary Fig. 14). In the first stage, the $\lambda_{em}$ showed a redshift from 463–485 nm, and the PL intensity was enhanced only two times which was smaller than that of MQ and SQ. A dramatic redshift of $\lambda_{em}$ (from 485–533 nm) was also observed for MeOSQ in the second stage, which was caused by the intermolecular interactions as evidenced by its crystal structure (Supplementary Fig. 18). As a result, MeOSQ showed a similar AIE effect ($I_{90}/I_0 = 7$) with SQ.

DPASQ, serving as the strongest D–A structure, exhibited a different trend of PL spectra (Fig. 2d). With the increase of $f_w$, the PL intensity first decreased with redshifted $\lambda_{em}$, which should be attributed to the typical TICT effect (Fig. 2e)[28,49,50]. When $f_w > 60\%$, it showed typical AIE properties with enhanced intensity due to the formation of aggregates and RIM, although the PL intensity at $f_w = 90\%$ was still lower than that in pure solution. Meanwhile, the blue-shifted $\lambda_{em}$ should be ascribed to two reasons. The first one is the decreased polarity inside the formed aggregates[51,52], and the other is the twisted conformation which is adverse to the intermolecular π–π interaction[53]. The fluorescent spectra and PL intensity change of DPASQ in MeOH/water also supported its TICT and AIE features (Supplementary Fig. 20). In addition, its crystalline sample with close packing showed a bright yellow light with a $\Phi$ of 10.2%, which was higher than the other three compounds (Supplementary Figs. 15, 19).

A summary of the photophysical properties of these four AIEgens was drawn in Table 1. In the first stage, the PL properties were mainly

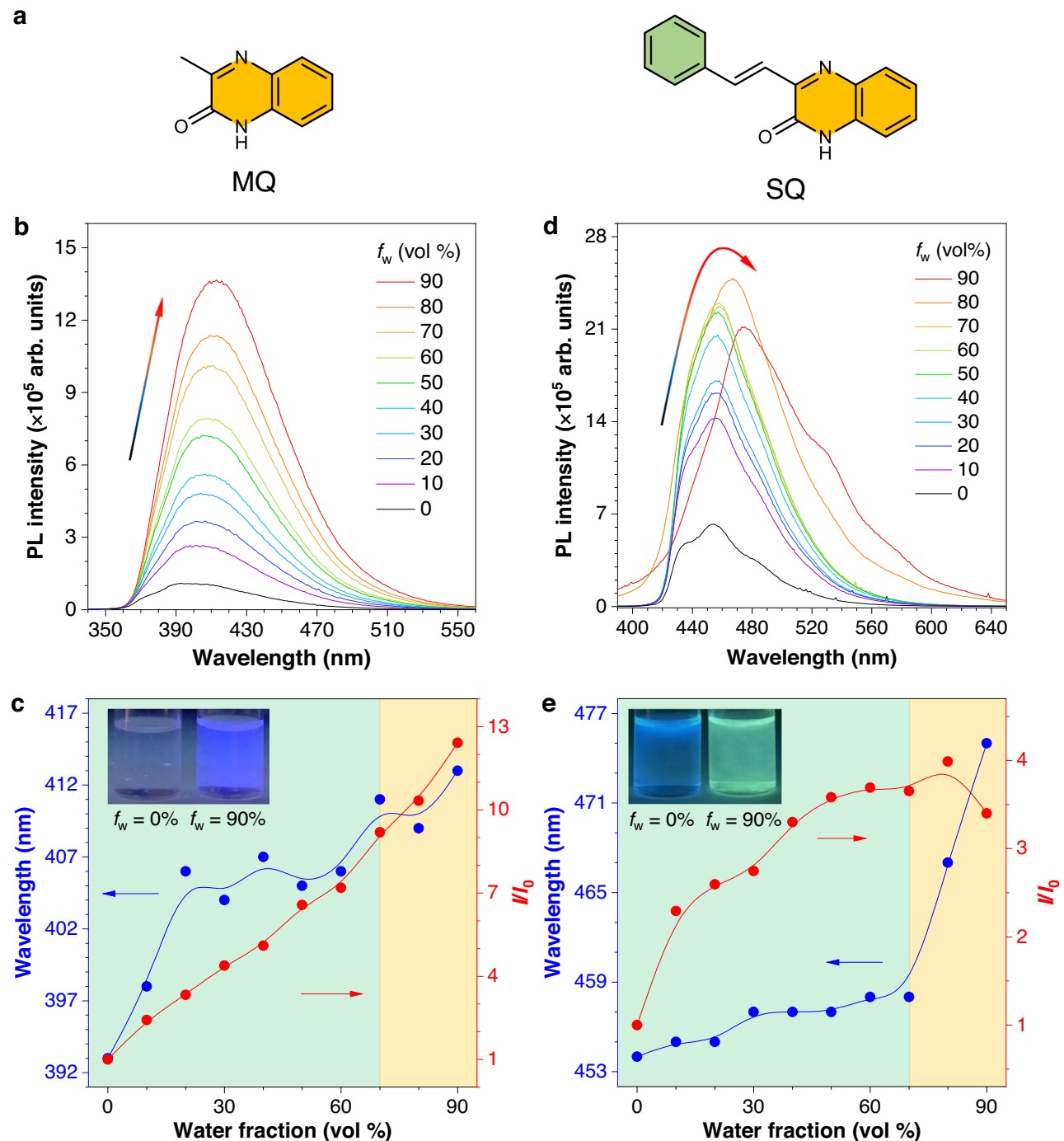

**Fig. 1 | Photophysical properties of MQ and SQ. a** The chemical structures of MQ and SQ. **b** Photoluminescence (PL) spectra of MQ in THF/water mixtures with different water fractions ($f_w$). Concentration ($c$) = $10^{-5}$ M, excitation wavelength ($\lambda_{ex}$) = 320 nm. **c** Plots of relative PL intensity ($I/I_0$) and maximum emission wavelength versus $f_w$. $I_0$ = PL intensity at $f_w$ = 0%. Inset: fluorescent photographs of MQ in the THF/water mixtures with $f_w$ of 0% and 90%, respectively. **d** PL spectra of SQ in THF/water mixtures with different $f_w$. $c$ = $10^{-5}$ M, $\lambda_{ex}$ = 350 nm. **e** Plots of $I/I_0$ and maximum emission wavelength versus $f_w$. Inset: fluorescent photographs of SQ in the THF/water mixtures with $f_w$ of 0% and 90%, respectively.

affected by the solvent polarity. From MQ to SQ, MeOSQ, and DPASQ, with the increase of electron-donating ability, the PL intensity enhancement was gradually weakened, and DPASQ even showed a decreased emission intensity. However, the redshifted $\lambda_{em}$ was more obvious in the compounds with a strong donor. For the second stage of aggregation, all of these four molecules exhibited the AIE effect, but the change of $\lambda_{em}$ and PL intensity was synergistically controlled by their aggregate structures and environmental polarity, resulting in complicated photophysical properties. In the crystalline state, they

displayed varying $\Phi$ (from 2.1% to 10.2%) but the same nature of fluorescence as suggested by their nanosecond lifetimes (Supplementary Fig. 21).

According to previous reports, the photophysical properties of these four heterocycle-containing AIEgens in the first stage should be closely related to PE[30,44,46]. In other words, the PE is successfully regulated by altering the electron-donating group of AIEgens and solvent environment in this work. To further verify such a conclusion, the PL spectra of these four compounds were

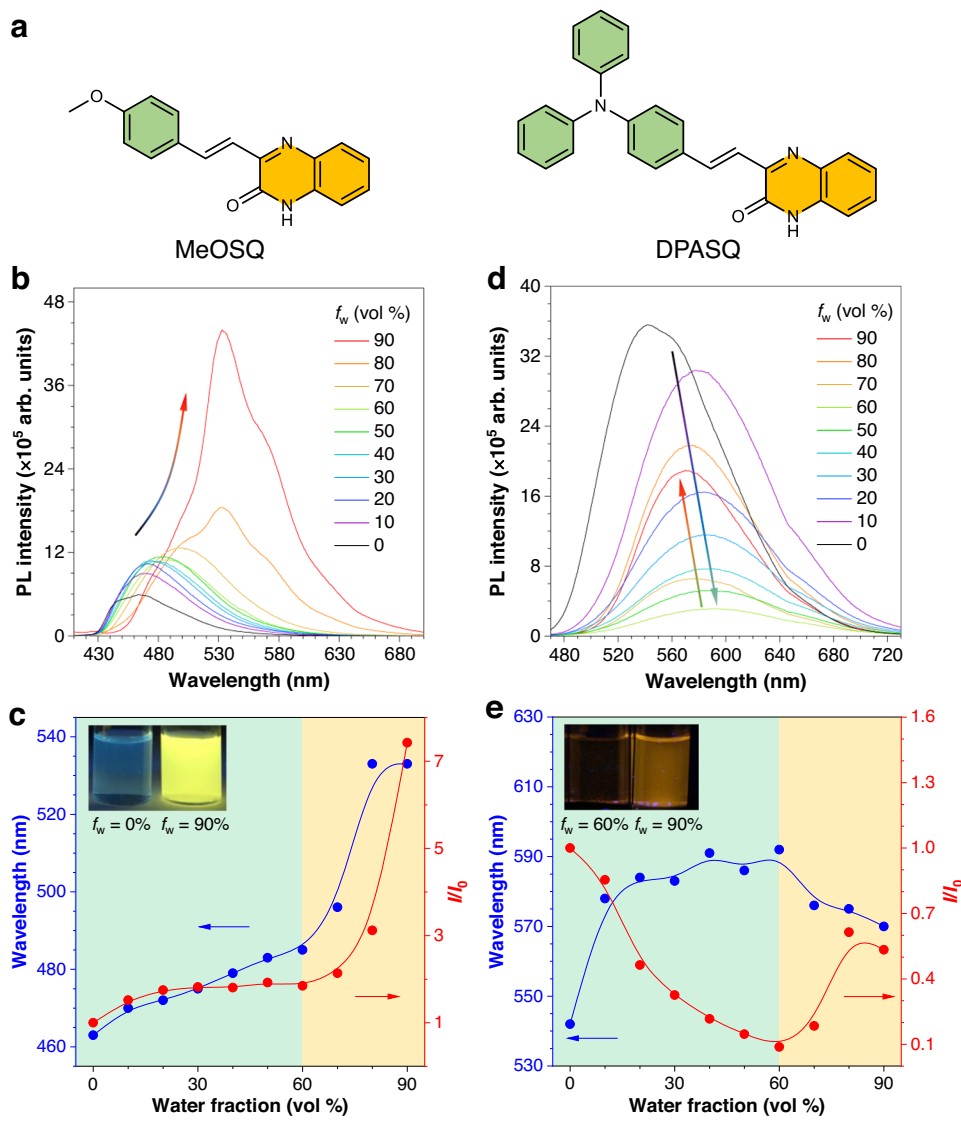

**Fig. 2 | Photophysical properties of MeOSQ and DPASQ. a** The chemical structures of MeOSQ and DPASQ. **b** Photoluminescence (PL) spectra of MeOSQ in THF/water mixtures with different water fractions ($f_w$). Concentration ($c$) = $10^{-5}$ M, excitation wavelength ($\lambda_{ex}$) = 390 nm. **c** Plots of relative PL intensity ($I/I_0$) and maximum emission wavelength versus $f_w$. $I_0$ = PL intensity at $f_w$ = 0%. Inset: fluorescent photographs of MeOSQ in the THF/water mixtures with $f_w$ of 0% and 90%, respectively. **d** PL spectra of DPASQ in THF/water mixtures with different $f_w$. $c$ = $10^{-5}$ M, $\lambda_{ex}$ = 430 nm. **e** Plots of $I/I_0$ and maximum emission wavelength versus $f_w$. Inset: fluorescent photographs of DPASQ in the THF/water mixtures with $f_w$ of 60% and 90%, respectively.

also measured and compared in different solvents with varying polarities (Fig. 3)[54,55]. From nonpolar hexane to polar methanol, the solvation effect was relatively weak in MQ and SQ which possessed no or a weak donor. However, in strong D−A structures of MeOSQ and DPASQ, an obvious redshift of $\lambda_{em}$ was observed with the increased solvent polarity. Especially, the $\lambda_{em}$ of DPASQ shifted from 480–600 nm (Fig. 3d). More importantly, four compounds showed different changes in PL intensity along with the increase of solvent polarity (Fig. 3e–h, Supplementary Fig. 22 and Supplementary Table 2). MQ displayed monotonically increased PL intensity with a maximum $I/I_{min}$ of 5, and SQ showed a similar enhancement with a smaller slope (the maximum $I/I_{min}$ =1.9). A similar phenomenon was also observed in some pyrene aldehyde systems[56–58]. However, the change is complex for MeOSQ without a clear trend. DPASQ even exhibited first increased and rapidly weakened PL intensity, suggesting its typical TICT effect that stabilized in polar solvents. In addition, although the tautomeric balance and monomer-dimer equilibrium of pyridines were well known, the same photophysical properties of methyl-substituted MQ (MeMQ) and unchanged $^{13}$C NMR spectra of MQ in different solvents strongly ruled out influence from these possibilities (Supplementary Figs. 23, 24)[59–62]. Therefore, the above results proved that PL properties in the first stage are mainly determined by the polarity and PE.

**Table 1 | Comparison of the photophysical properties for MQ, SQ, MeOSQ, and DPASQ toward water addition in THF/water mixtures$^a$**

| AIEgen | The first stage (polarity) | | | The second stage (aggregation) | | |
|---|---|---|---|---|---|---|
| | $f_w$ (%) | $\lambda_{em}$ (nm) | $I/I_0$ | $f_w$ (%) | $\lambda_{em}$ (nm) | $I/I_0$ |
| MQ | 0–70 | 393–410 | 9 | 70–90 | 410–413 | 13 |
| SQ | 0–70 | 454–458 | 4 | 70–90 | 458–475 | 4 |
| MeOSQ | 0–60 | 463–485 | 2 | 60–90 | 485–533 | 7 |
| DPASQ | 0–60 | 542–592 | 0.1 | 60–90 | 592–570 | 0.6 |

$^a f_w$ water fraction, $\lambda_{em}$ maximum emission wavelength in THF/water mixtures, $I/I_0$ relative PL intensity, $I_0$ PL intensity at $f_w$ of 0%.

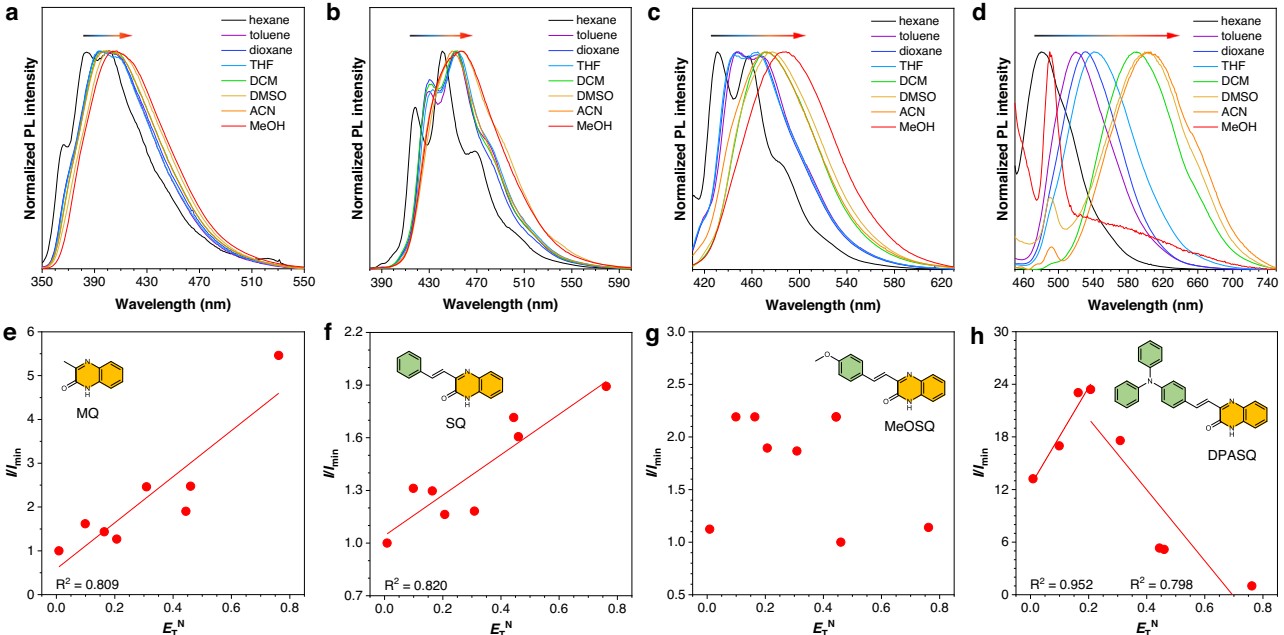

**Fig. 3 | Photoluminescence (PL) properties of MQ, SQ, MeOSQ, and DPASQ in different solvents. a–d** Normalized PL spectra of **a** MQ, **b** SQ, **c** MeOSQ, and **d** DPASQ in different solvents with increased polarity (*n*-hexane (hexane) < toluene < dioxane < tetrahydrofuran (THF) < dichloromethane (DCM) < dimethyl sulfoxide (DMSO) < acetonitrile (ACN) < methanol (MeOH)). Concentration = $10^{-5}$ M. The arrow indicates the increase of solvent polarity. **e–h** The relative PL intensity ($I/I_{min}$) of **e** MQ, **f** SQ, **g** MeOSQ, and **h** DPASQ in different solvents versus the normalized Reichardt's parameter ($E_T^N$). $I_{min}$ = the lowest PL intensity of the target compound in different solvents. The linear fitting (red line) represents the trend of $I/I_{min}$ in different solvents with increased polarity, $R^2$ = goodness of fit.

## Theoretical analysis

To investigate the mechanism behind the different photophysical properties of the four compounds, theoretical analysis based on time-dependent density functional theory (TD-DFT) was first applied. Generally, in heteroatom-containing compounds, the ($n,\pi^*$) state with small oscillator strength (*f*) is regarded as the nonradiative decay channel that may quench the emission. In contrast, the ($\pi,\pi^*$) state with a large value of *f* is considered an emissive state (Supplementary Fig. 25)[63–66]. Therefore, the adiabatic energy levels and corresponding hole-electron structures of the lowest-lying ($n,\pi^*$) and ($\pi,\pi^*$) states in different solvents were calculated and analyzed for these four compounds based on their optimized ground- and excited-state geometries (Fig. 4 and Supplementary Tables 3–6). The hole-electron analysis showed a distinct difference between the two kinds of electron transition. The heteroatoms of the methylquinoxaline unit mainly dominated the ($n,\pi^*$) state with an *f* value close to zero, while π-conjugated donor and heterocyclic skeleton participated in the emissive ($\pi,\pi^*$) state. The *f* value of ($\pi,\pi^*$) transition gradually became larger with the increased polarity of solvents and the electron-donating ability of donors.

For MQ, the energy level of the nonemissive ($n,\pi^*$) state was lower than that of the emissive ($\pi,\pi^*$) state. However, with the increase of solvent polarity, the energy levels of ($n,\pi^*$) and ($\pi,\pi^*$) states were increased and decreased, respectively. As a result, the energy gap between these two states became smaller and almost the same in polar solvents (e.g., MeOH). Meanwhile, the gradually enhanced vibronic coupling of ($n,\pi^*$) and ($\pi,\pi^*$) states induced the PE and boosted the radiative decay from ($\pi,\pi^*$) state, which was the reason why MQ showed PL intensity enhancement. The decreased energy level of ($\pi,\pi^*$) state also resulted in the redshift of the enhanced emission. Interestingly, according to previous work, this should be an anti-Kasha's emission from the higher excited state due to the thermal equilibrium between two coupled states and a much larger *f* value (>0.21) of ($\pi,\pi^*$) state[67–69]. However, for SQ, the energy level of ($\pi,\pi^*$) state was lower than ($n,\pi^*$) state, which changed from a closely

coupled feature to the ($\pi,\pi^*$) dominant lowest state with the increased solvent polarity. Although PE helped MQ boost the PL intensity, it rendered SQ a small enhancement of PL intensity because the bright ($\pi,\pi^*$) state always showed a lower energy level than the ($n,\pi^*$) state.

For MeOSQ and DPASQ with strong donor groups, the energy level of the bright ($\pi,\pi^*$) state was always lower than their corresponding ($n,\pi^*$) state in all solvents. The energy gaps between these two states became larger with the increased polarity, and the gap of DPASQ was bigger than MeOSQ in the same solvent. Due to the negligible PE on them, theoretically, these two compounds should exhibit a slight enhancement of PL intensity with the increased solvent polarity. However, apart from PE, it needs to be emphasized that TICT is also a dominant effect in strong D−A systems with a rotatable bond, which always induces decreased PL intensity and redshifted $\lambda_{em}$ with the increase of solvent polarity. Therefore, PE, TICT, and RIM work synergistically to affect the photophysical properties of MeOSQ and DPASQ in the solvent mixtures, especially for DPASQ that showed a "V" shape of PL intensity variation and dramatically redshifted $\lambda_{em}$ with the increased water fraction (Fig. 2e).

## Regulation of proximity effect

Accordingly, an ideal model of regulating PE and photophysical properties in these heterocycle-containing AIE systems by introducing π−conjugated donors was summarized (Fig. 5). From the methylquinoxaline core without a donor to three derivatives with donors and increased electron-donating ability, the energy levels of ($n,\pi^*$) and ($\pi,\pi^*$) states gradually increased and decreased, respectively (Fig. 5a). Thereinto, the energy-level crossing of ($n,\pi^*$) and ($\pi,\pi^*$) states occurred, and vibronic coupling and PE between these two states were firstly enhanced but further weakened. As a result, these four AIEgens exhibited different photophysical properties during the first stage of adding water to the THF solution (Fig. 5b−e): (1) When the nonemissive ($n,\pi^*$) dominated the lowest-lying state, the addition of water would narrow the gap and enhance the vibronic coupling between ($n,\pi^*$) and

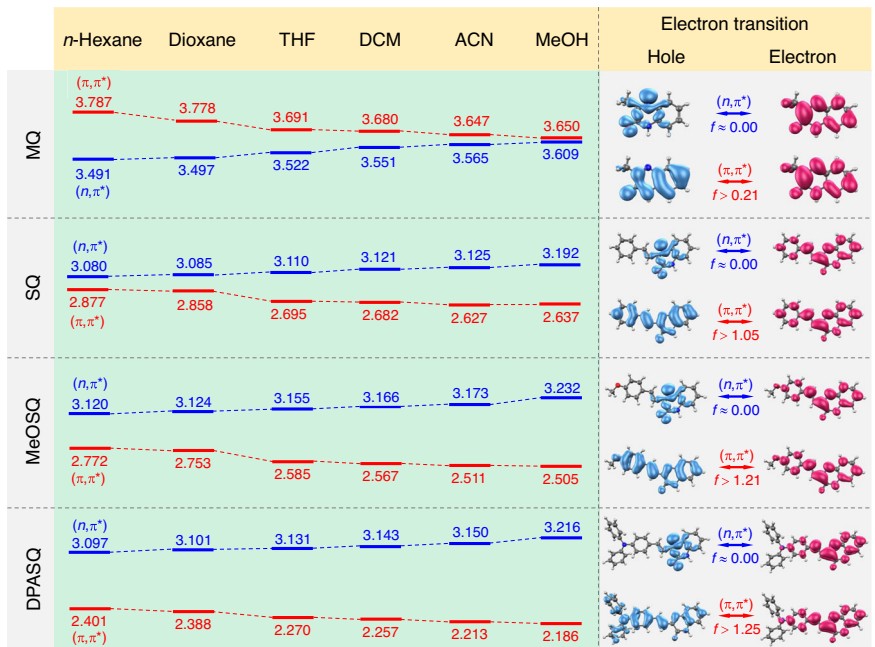

**Fig. 4 | Adiabatic energy levels and hole-electron analysis.** Left panel: the adiabatic energy levels of the lowest-lying $(n,\pi^*)$ and $(\pi,\pi^*)$ states in different solvents based on their optimized ground- and excited-state geometries, calculated using TD-DFT method at the theory level of PBE0-D3/6−31 G(d,p), Gaussian 16. Unit = eV. Right panel: the hole-electron analysis and oscillator strength (*f*) of $(n,\pi^*)$ and $(\pi,\pi^*)$ states.

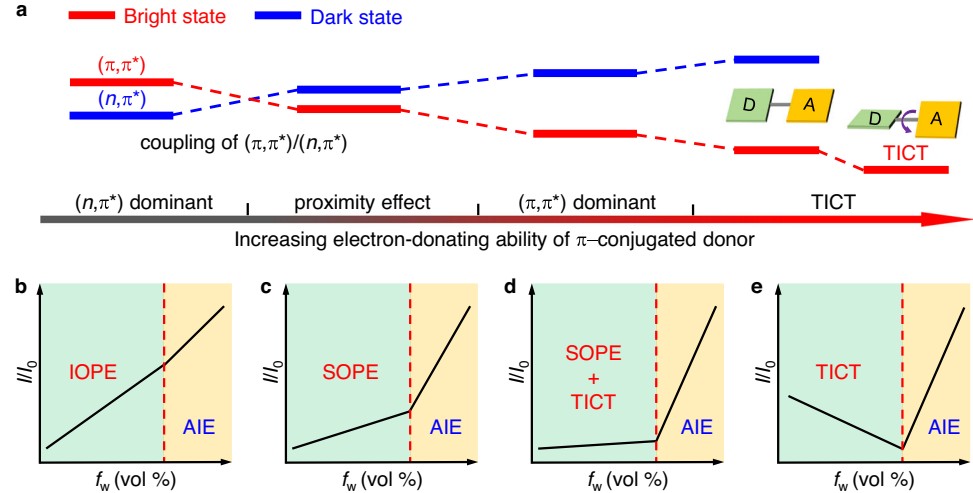

**Fig. 5 | Schematic illustration of regulating proximity effect. a** The energy levels and $(n,\pi^*)/(\pi,\pi^*)$ coupling of the proximity effect with the increased electron-donating ability of π−conjugated donors. **b**–**e** Schematic plots of relative photoluminescence intensity in THF/water mixtures for AIEgens with different electronic structures. AIE aggregation-induced emission, IOPE induction of proximity effect, SOPE suppression of proximity effect, and TICT twisted intramolecular charge transfer.

$(\pi,\pi^*)$ states, which led to radiative decay from the emissive $(\pi,\pi^*)$ state and enhanced PL intensity (Fig. 5b). Accordingly, this phenomenon typically shown by MQ is termed the induction of proximity effect (IOPE), which indicates that PE is not always a negative influence as reported before. (2) Introducing a phenyl group resulted in the closely coupled two states of SQ. Therefore, the $(\pi,\pi^*)$ state became the dominant state, and the decoupling of the two close-lying states happened, resulting in the suppression of proximity effect (SOPE) and increased PL intensity (Fig. 5c). (3) Further increasing the electron-donating ability of the donor, a weak TICT effect was introduced into the system. Meanwhile, since the energy gap between $(\pi,\pi^*)$ and $(n,\pi^*)$ became larger, the strength of PE dramatically declined. Hence, these two opposite effects endow MeOSQ with a net effect of very weak intensity enhancement (Fig. 5d). (4) When the electron-donating ability of the π−conjugated donor was further strengthened, the TICT effect was obvious, and the PE disappeared due to the very large energy gap. Accordingly, TICT became the key mechanism to influence photophysical properties, which endowed DPASQ with gradually decreased PL intensity and dramatically redshifted $\lambda_{em}$ toward water addition (Fig. 5e). Nevertheless, in the second stage of adding water to the THF solution ($f_w \geq 60$ for MQ and SQ, and $f_w \geq 70$ for MeOSQ and DPASQ), four AIEgens displayed similar AIE phenomenon with gradually enhanced PL intensity after aggregation. The above mechanistic perspective suggests the successful manipulation of PE and photophysical properties for heterocycle-containing AIE systems with close energy levels of $(n,\pi^*)$ and $(\pi,\pi^*)$ states.

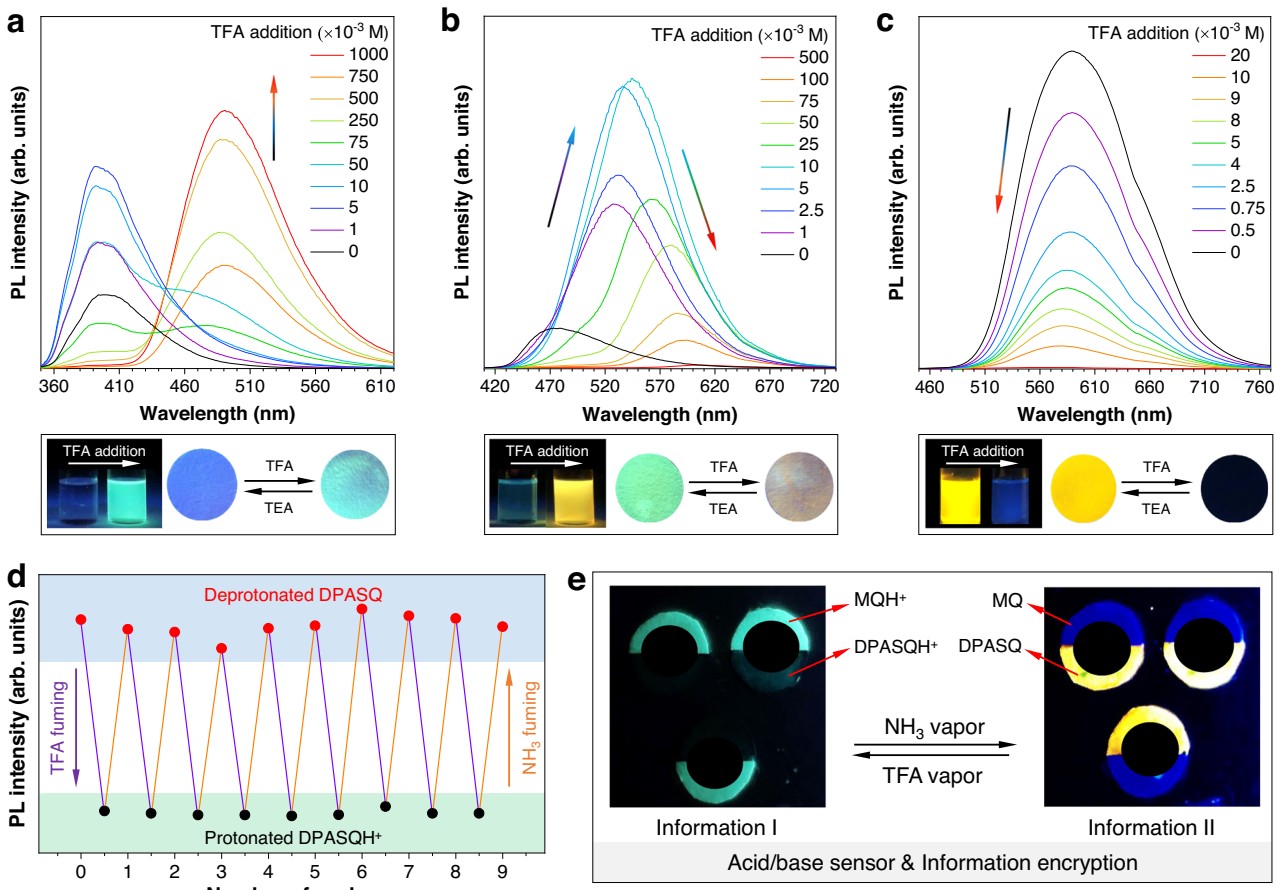

**Fig. 6 | Dynamic responses to acid/based stimuli and applications. a–c** Top panel: photoluminescence spectra of **a** MQ, **b** MeOSQ, and **c** DPASQ solutions in dichloromethane upon adding trifluoroacetic acid (TFA). The concentration of luminogens is $10^{-4}$ M. Bottom panel: fluorescent photographs of responses towards protonation by addition of TFA to solutions or fuming films deposited on filter paper by TFA gas. TEA triethylamine. **d** Fluorescent intensity of protonated and deprotonated DPASQ films upon sequential NH$_3$ and TFA fuming. **e** Photographs of fluorescent responses of MQ and DPASQ upon NH$_3$ and TFA exposure for sensor and information encryption.

## Dynamically responsive behavior

The intrinsic feature of heteroatoms always endows heterocycle-containing compounds with responses to protonation, which could be utilized as chemical sensors for advanced applications. Therefore, taking MQ, MeOSQ, and DPASQ as examples, their dynamic responses to protonation were further investigated (Fig. 6a–c). After adding trifluoroacetic acid (TFA) liquid into the MQ solution, a new red-shifted absorption peak at 380 nm was gradually enhanced, which was supported by theoretical calculation (Supplementary Figs. 26a, 27a). Its PL intensity at 397 nm was first enhanced due to the increased polarity of the mixture and the resulting IOPE effect (Fig. 6a). Then, the original emission peak dropped, and a new peak at 500 nm increased. As a result, protonation gave rise to an enhanced and redshifted emission. In addition, the original absorption peaks of MeOSQ and DPASQ gradually decreased together with the rise of a new redshifted absorption peak (i.e., 518 nm for MeOSQ and 640 nm for DPASQ, respectively) due to nitrogen protonation of the methylquinoxaline core (Supplementary Figs. 26, 27). Although the absorption changes were quite similar, they displayed different PL properties. The fluorescence of MeOSQ was first raised and red-shifted due to both the SOPE effect and enhanced CT of the proto-nated compound (Fig. 6b). However, as the protonation progressed, its fluorescence was redshifted but weakened because of the man-ifest self-absorption. Therefore, the protonation of MeOSQ brought about a redshifted emission with nonmonotonically changed inten-sity. In contrast, the original emission peak of DPASQ at 582 nm

gradually decreased with the addition of TFA (Fig. 6c). The proto-nated DPASQH$^+$ showed an enhanced TICT effect and a much lower transition possibility ($f = 0.177$) than DPASQ ($f = 1.541$), supporting its nonemissive nature and the declined PL intensity (Supplementary Fig. 28). Besides, no protonation of the electron-donor unit or double protonated compound was observed due to the poor basic ability of triphenylamine (Supplementary Fig. 27c)[70]. As a result, the protona-tion of DPASQ produced a monotonically weakened emission, which was an opposite phenomenon to MQ.

For compounds that are responsive to acid stimuli, their pre-acidified forms are usually responsive to basic stimuli (e.g., NH$_3$) as well. Taking DPASQ as an example, its protonation and deprotonation processes were continuously studied, indicating the nature of excel-lent reversibility and repeatability (Fig. 6d). Therefore, dynamic responses endow these compounds with smart sensors or information encryption applications. As a demonstration, a reusable test stripe based on MQ and DPASQ with different changes was fabricated to realize the above functions (Fig. 6e). Pattern transformations were obtained upon repeated NH$_3$ and TFA exposure with the help of their opposite turn-on and turn-off modes. As a result, one mode of the test stripe with protonated compounds (MQH$^+$ and DPASQH$^+$) showed a smiling face which could be translated as "Information I". Another mode with deprotonated compounds (MQ and DPASQ) displayed an angry face regarded as "Information II". These two pieces of informa-tion are easily controllable by changing the external acid/base environment.

## Discussion

PE plays an essential role in controlling the photophysical properties of heterocycle-containing compounds with close energy levels of the lowest $(\pi,\pi^*)$ and $(n,\pi^*)$ states, which is usually regarded as a non-radiative decay channel for luminescent materials. In this work, several heterocyclic methylquinoxaline derivatives with different electron-donating groups were synthesized to study the regulation of PE. All these compounds were verified as luminogens with AIE properties, but they exhibited different photophysical properties and AIE curves toward water addition during the aggregation process. The parent core of MQ without a donor showed $(n,\pi^*)$ dominant low-lying state and IOPE, while SQ with a weak donor displayed strong vibronic coupling between $(n,\pi^*)$ and $(\pi,\pi^*)$ states and SOPE during water addition. MeOSQ with a moderate electron donor showed $(\pi,\pi^*)$ dominant low-lying state and weak SOPE, but the opposite TICT effect dominated DPASQ due to its strongest electron donor. The theoretical analysis provided a mechanistic perspective on energy level change and PE regulation of these AIEgens, suggesting both the positive and negative impacts of PE for luminogens with different electronic states, which overturned our solidified recognition of PE. As a demonstration, these AIEgens were successfully applied as sensors with dynamic responses to acid/base stimuli based on the intrinsic nature of heterocycles and PE. This work provides a strategy to regulate the frequently ignored PE in the AIE community and also realizes the controllable molecular design of heterocyclic AIEgens for advanced applications with dynamic responses.

## Methods

### Materials

3-methylquinoxalin-2(1H)-one (MQ, 95%), benzaldehyde (> 98.0%), 4-methoxybenzaldehyde (> 99.0%), 4-(diphenylamino)benzaldehyde (> 98.0%) were purchased from TCI (Shanghai) Development Co., Ltd. MQ was purified using column chromatography with dichloromethane, and other chemicals were utilized as received without further purification. Dimethyl sulfoxide-$d_6$, used for nuclear magnetic resonance (NMR) measurements, was purchased from Energy Chemical. Tetrahydrofuran (THF) used in experiments was distilled from sodium benzophenone ketyl under nitrogen gas.

### Instrumentation

NMR measurements of $^1$H and $^{13}$C were conducted on a Bruker AVIII 400 MHz NMR spectrometer equipped with a Dual Probe. High-resolution mass spectra (HRMS) were performed on a CT Premier CAB048 mass spectrometer using a positive ESI-TOF module. Ultraviolet-visible (UV-Vis) spectra were recorded on a Varian Cary 50 UV-Visible Spectrophotometer. Photoluminescence (PL) spectra were carried out on Fluorolog®-3 (HORIBA) spectrofluorometer. Absolute quantum yields (QY) were collected using an integrating sphere on a Hamamatsu Quantum Yield Spectrometer C11347 Quantaurus. Fluorescence lifetime was recorded on an Edinburgh FLS980 Spectrometer. Dynamic light scattering (DLS) of the diameters of aggregates was measured on Malvern Zetasizer Nano ZS equipment at room temperature. Single-crystal structures of four compounds were confirmed through single-crystal X-ray diffraction on a Rigaku Oxford Diffraction (SuperNova) with Atlas diffractometer (Cu K$\alpha$ ($\lambda$ = 1.54184 Å) and solved using Olex2 software. All digital photographs of the solution, mixture, and crystalline powder were recorded using a Canon EOS 60D camera.

### Computational details

The geometries of all compounds in the ground state were optimized using the density functional theory (DFT) method at the PBE0/6-31 G(d,p) level with Grimme's DFT-D3 correction, which was widely utilized to evaluate luminescent materials and their energy levels with comparatively high accuracy[71–74]. The excited-state geometries were optimized using the time-dependent DFT method at the same level of theory. To investigate the solvent effect on the energy level and electron transition of these compounds, the solvation model based on the density model of SMD and self-consistent reaction field was considered in the calculations. The adiabatic and vertical energy levels were summarized based on their optimized ground- and excited-state geometries. All the above calculations were carried out using Gaussian 16 program (Revision A.03). The frontier molecular orbitals were displayed using the IQmol molecular viewer package (Version 3.0.1).

## Data availability

The authors declare that all the data supporting the findings of this manuscript are available within the manuscript and Supplementary Information files and available from the corresponding authors upon request. The X-ray crystallographic coordinates for structures reported in this study have been deposited at the Cambridge Crystallographic Data Centre (CCDC) under deposition numbers 2213287 (MQ), 2213288 (SQ), 2213289 (MeOSQ), and 2213290 (DPASQ). These data can be obtained free of charge from The Cambridge Crystallographic Data Centre via www.ccdc.cam.ac.uk/data_request/cif.

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

## Acknowledgements

This work was supported by the National Natural Science Foundation of China Grant (21788102, 22205197), the Research Grants Council of Hong Kong (16307020, C6014-20W, N_HKUST609/19 and 16305320), the Innovation and Technology Commission (ITC-CNERC14SC01), the Natural Science Foundation of Guangdong Province (2019B121205002 and 2019B030301003), Shenzhen Key Laboratory of Functional Aggregate Materials (ZDSYS20211021111400001), and the Science Technology Innovation Commission of Shenzhen Municipality (KQTD20210811090142053 and JCYJ20220818103007014). J. Zhang acknowledged the support from the Research Grants Council Postdoctoral Fellowship Scheme of Hong Kong Special Administrative Region, China (HKUST PDFS2324-6S01).

## Author contributions

J.Z., Y.T., and B.Z.T. conceived and designed the experiments. J.Z. and Y.T. performed the synthesis. J.Z., Y.T., and H.S. conducted the photophysical measurements and theoretical calculations and together analyzed the data. J.Z., J.W.Y.L., J.S., H.Z., and B.Z.T. participated in the discussion and gave essential suggestions. J.Z., H.Z., and B.Z.T. co-wrote and revised the paper.

## Competing interests

The authors declare no competing interests.
