## [Peer Review File · Nature Communications]

Regulating the proximity effect of heterocycle-containing
AIEgensREVIEWER COMMENTS

Reviewer #1 (Remarks to the Author):

Zhang, Tang et al. describe the luminescence properties of a series of methylquinoxaline derivatives, rationalising their observations by invoking the 'proximity effect', i.e., situations where the n,π^* and π,π^* electronically excited states are close in energy to one another. They conclude that the ordering of these states and the size of the energy gap between them dictate the extent of emission wavelength changes and emission intensity enhancement that occurs when the materials are dissolved in solvents of different polarities or when they form aggregates. Previously, the proximity effect has not been considered in the design of aggregation-induced emission chromophores and it has generally been thought of as a deleterious effect. The results in the current manuscript apparently suggest that there are cases where the proximity effect can be manipulated to improve aggregation-induced emission properties.

In principle, I believe that these observations are novel and of sufficient importance to warrant publication in Nature Communications. However, there are major concerns that the authors should address before I would have full confidence in their mechanistic description (described below as points 1 and 2). To me, there are other plausible explanations other than the proximity effect that could account for the observed emission properties, and these would have to be excluded through control experiments.

1. Like pyridines, the methylquinoxaline structure has the potential to form tautomers through protomeric equilibrium – a keto form, as drawn in the manuscript, and an aromatic form. It is well known for pyridones that the balance of these tautomers is influenced by changes in solvent polarity (e.g., see J. Chem. Soc. Perkin Trans. 2, 1976, 12, 1428). Based on this literature, one might expect that the aromatic form of the methylquinoxaline would be favoured in nonpolar solvents, but the keto form would be favoured in polar solution. A shift in this equilibrium could be responsible for the observed changes in emission profile. Therefore, it is surprising that this possible equilibrium does not seem to be mentioned at all in the manuscript. Experiments should be carried out to investigate if the methylquinoxalines are prone to this equilibrium. From a brief look at the pyridone literature, it is not trivial to establish the nature of this type of equilibrium because the spectroscopic signals (e.g., the OH/NH ^1H resonances) are similar for the two tautomers and can be in rapid exchange. However, it is important to gain experimental insight into this process to understand if it is responsible for the observations, rather than the proximity effect.

2. Similarly, it is well known that pyridones can undergo dimerization through self-complementary hydrogen bonding. Changes in monomer–dimer equilibria would be another, alternative explanation for the observed changes in emission in the presence of different solvents which would have to be excluded before invoking the proximity effect. This could be achieved by repeating measurements at a series of different concentrations of the luminogen as the monomer–dimer equilibrium will be concentration dependent.

Some additional minor comments:

3. In the abstract, it would help to clarify that the proximity effect deals with states that are close 'in energy'. As written currently, it could imply that the states are close in space.
4. No yields are given for the synthetic procedures in the SI. The yields should be included.
5. Around lines 96-100, the authors mention that RIM effect influences the emission of MQ. What motions are being proposed as responsible for the nonradiative decay of MQ? It lacks the usual, rotatable aryl groups that are found commonly in AIE luminogens.
6. Information should be given about the red lines added to the data in Figures 3e-h. How have they been fitted? What are the chi squared values? It may be better not to show any line if there is not a clear/unambiguous trend, e.g., especially for panel g.
7. Is there an explanation that can be given as to why the π,π^* states are stabilised by increased solvent polarity while the n,π^* states are destabilised?
8. Is there a reason that the authors have chosen the PBE0-D3/6-31G(d,p) method for TD-DFT? Has it been benchmarked to the experimental data, or have the authors screened other functionals and basis sets? Coupled cluster or ADC methods would likely give more reliable results for the excited state energy levels and are feasible for the small molecules being investigated. This point is particularly important given that calculated energies are a key component of the proposed 'induced' and 'suppressed' proximity effects.
9. On line 223, the authors imply that the convergence of the π,π^* and n,π^* energy levels causes a redshift in the emission. It's not immediately obvious why that would be. Can more explanation be given? Is it not the case that the redshift is caused by the greater solvent dipole relaxation in more polar solvents, independent of the spacing of these two energy levels relative to one another?
10. The amount of TFA in Figure 6 would be better given as a concentration rather than a volume. The volume of the DCM solution of the luminogen is not given, so there is no way for the reader to know the relative stoichiometry/concentration of the TFA relative to the luminogen. Stating the concentration of TFA avoids this problem and is the most informative way to report it for the reader.
11. The absorbance data in SI Figure 19 does not appear to show an isospeptic point for the titration of TFA into the solutions of MQ, MeOSQ and DPASQ, which is not consistent with a simple equilibrium from unprotonated to protonated luminogen. It may be more consistent with the presence of tautomers and/or dimers. This data should be considered in response to points 1 and 2.

Reviewer #2 (Remarks to the Author):

Comments:

In this manuscript, Tang et al. reported an electron-donor engineering strategy to manipulate the proximity effect (PE) of four methylquinoxaline (MQ) derivatives and revealed the impact of PE on AIEgens. Spectroscopic analysis was investigated to show their fluorescent change toward water addition and aggregation, and theoretical calculations systematically illustrated their electronic structures and energy-level variation of (n,π^*) and (π,π^*) states in solvent environments with different polarities. In addition, they studied the responses of these AIEgens to acid/base stimuli and demonstrated potential applications. Generally, PE has been regarded as the nonradiative decay channel for luminogens for many years, but it is interesting and significant that the authors revealed both the positive and negative impacts of PE for AIEgens with different electronic states. The present manuscript is an important contribution to AIE research as it (i) connects PE and AIE for the first time and (ii) provides a new perspective to design and regulate the photophysical properties of heterocycle-containing AIEgens. The work is innovative, well-justified, and generally well-written. Therefore, this manuscript is recommended for publication subject to the following minor issues.

1. Restriction of intramolecular motion (RIM) is proved as the key mechanism for AIE effect. However, in the introduction, the authors claimed that the combination of PE and RIM could replenish AIE mechanisms. Please explain the linkage between PE and RIM and how PE can provide new insights into the AIE mechanism.
2. It's reasonable that DPASQ exhibits the twisted intramolecular charge transfer effect (TICT) toward water addition (Fig. 2), while MeOSQ was different due to its weak D-A interaction. The author also provided their PL spectra in different solvents and conducted theoretical calculations to support the TICT effect. However, I am curious about their fluorescence change and AIE curve in polar solvents, e.g., DMSO/H₂O or MeOH/ H₂O mixtures.
3. As illustrated in Fig 4, the (n,π^*) state was always lower than (π,π^*) state of MQ, resulting in the anti-Kasha's emission. However, achieving such kind of emission is a difficult issue and requires strict conditions. The red-shifted emission of MQ in THF/ H₂O seems to support this anti-Kasha's behavior. Are there other reasons why the emission from MQ came from the higher (π,π^*) state instead of the (n,π^*) state?
4. The first stage behaviors of these four compounds toward water addition are carefully shown in Fig. 5. On the other hand, a simple description of their PL change after aggregation should be added as a complete illustration, although they show similar behavior.
5. Please explain why MeOSQ and DPASQ theoretically exhibit a slight enhancement of PL intensity and red-shifted emission with increased solvent polarity, as described on page 9.
6. Please list Reichardt's parameters and normalized parameters for each solvent used in Fig. 3.
7. Regarding the Methods part, the authors claimed that "all chemicals were utilized as received without purification". However, "MQ was purified with column chromatography", as stated in SI. Please check and unify the description.

8. There are several typos:

(i) Page 2, "aggregation-induced emission (AIE), ..., which referred to the luminogens". AIE should refer to an effect or phenomenon instead of luminogens.

(ii) Page 5, "typical" should be "typically".

Reviewer #3 (Remarks to the Author):

Using four kinds of AIE compounds, the authors claim that the enhancement of luminescence intensity at the initial stage of aggregate formation can be controlled by solvent polarity based on the PE effect. They finally state that this mechanism can be applied to AIE in heterocycles. Although it is interesting as a proposal on the AIE mechanism, it is considered that there are few experimental examples to claim generality. Therefore, I judged that this manuscript is not suitable for publication at this stage. The following points must be considered.

-Emission mechanism of SQ is similar to that of pyrene aldehyde, while the others show emission from the CT states. It is an exaggeration to compare those with different emission mechanisms under the same theoretical framework, even though they all show AIE.

-Also, to insist generality of this effect in all heterocyclic systems, further examples are needed. At least, with pyrimidines which have different aza-substituted positions and heteroles which have different sizes, the authors need to show the similar effects.

-To discuss the solid-state luminescence properties in AIE behaviors, the morphology must be clarified. In particular, since molecules can aggregate or crystallize at the initial stage by increasing water contents in the aggregation experiments, comparisons between undefined structures can misinterpret results if morphology is not unified.

-What about the triplet state properties of your molecules? In general, the low-lying (n,π^*) state of carbonyl compounds is responsible for their fast intersystem crossing.

-In Figures 1b, 1d, 2b, 2d and 3, it is difficult to recognize the relationships between the peak shift and parameter changes. Please insert the arrow and rearrange line colors, such as rainbow color, to readily understand the increase of water contents or polarity.

Point-by-point Response to the Reviewers' Comments

Response to the Comments of Reviewer 1:

Zhang, Tang et al. describe the luminescence properties of a series of methylquinoxaline derivatives, rationalising their observations by invoking the 'proximity effect', i.e., situations where the n,π^* and π,π^* electronically excited states are close in energy to one another. They conclude that the ordering of these states and the size of the energy gap between them dictate the extent of emission wavelength changes and emission intensity enhancement that occurs when the materials are dissolved in solvents of different polarities or when they form aggregates. Previously, the proximity effect has not been considered in the design of aggregation-induced emission chromophores and it has generally been thought of as a deleterious effect. The results in the current manuscript apparently suggest that there are cases where the proximity effect can be manipulated to improve aggregation-induced emission properties.

In principle, I believe that these observations are novel and of sufficient importance to warrant publication in Nature Communications. However, there are major concerns that the authors should address before I would have full confidence in their mechanistic description (described below as points 1 and 2). To me, there are other plausible explanations other than the proximity effect that could account for the observed emission properties, and these would have to be excluded through control experiments.

Response: Thanks a lot for the reviewer's recognition of the novelty and importance of our work. According to the valuable suggestions from the reviewer, we have added some control experiments and carefully revised the manuscript. We hope that these supplementary experiments and answers can address your concerns.

1. Like pyridines, the methylquinoxaline structure has the potential to form tautomers through protomeric equilibrium – a keto form, as drawn in the manuscript, and an aromatic form. It is well known for pyridones that the balance of these tautomers is influenced by changes in solvent polarity (e.g., see J. Chem. Soc. Perkin Trans. 2, 1976, 12, 1428). Based on this literature, one might expect that the aromatic form of the methylquinoxaline would be favoured in nonpolar solvents, but the keto form would be favoured in polar solution. A shift in this equilibrium could be responsible for the observed changes in emission profile. Therefore, it is surprising that this possible equilibrium does not seem to be mentioned at all in the manuscript. Experiments should be carried out to investigate if the methylquinoxalines are prone to this equilibrium. From a brief look at the pyridone literature, it is not trivial to establish the nature of this type of equilibrium because the spectroscopic signals (e.g., the OH/NH 1H resonances) are similar for the two tautomers and can be in rapid exchange. However, it is important to gain experimental insight into this process to understand if it is responsible for the observations, rather than the proximity effect.

Response: Thanks a lot for your careful review and comments. As suggested by the reference, it is easy for pyridines to realize tautomerism between keto/enol forms (e.g., J. Chem. Soc. Perkin Trans. 2, 1976, 12, 1428). However, the balance of tautomers is largely affected by substituents bonded to the pyridine ring (*Arkivoc* 2002, 11, 198). For quinoxaline, the keto form usually dominates in both solution and solid phases, especially for quinoxalin-2-one (a derivative of MQ without methyl group) that physical measurements fail to demonstrate the presence of the hydroxy form (*Adv. Heterocycl. Chem.* 1963, 2, 203). Therefore, it is believed

that the keto form dominates the MQ structure in both solution and solid phases, and no obvious tautomerism should be observed to influence the photophysical properties of MQ in different solvents with different polarities.

To further support the above conclusion, two more control experiments were conducted using MQ as an example. (1) By replacing the hydrogen atom with a methyl group, the methyl-substituted MQ (namely MeMQ) was synthesized to rule out the effect of tautomerism (Supplementary Figures 2, 7, and 12). Its photophysical properties were measured and compared with MQ (Supplementary Figure 23). MeMQ and MQ showed almost the same PL spectra in THF/water mixtures and different solvents with different polarities, indicating their similar photophysical properties. Besides, the theoretical calculation also supports the same adiabatic energy levels and hole/electron distribution of MeMQ and MQ. Hence, these results strongly support that the tautomerism of MQ plays a negligible role in the observed changes in emission profile.

Supplementary Figure 23. The photophysical properties of methyl-substituted MQ (MeMQ). **a** The normalized absorption spectra in different solvents. Concentration = 10^{-5} M. **b** Photoluminescence (PL) spectra of MeMQ in THF/water mixtures with different water fractions (f_w). Concentration (c) = 10^{-5} M, excitation wavelength (λ_{ex}) = 320 nm. **c** Plots of relative PL intensity (I/I_0) and maximum emission wavelength versus f_w . I_0 = PL intensity at f_w

= 0%. $c = 10^{-5}$ M, $\lambda_{\text{ex}} = 320$ nm. **d** The normalized PL spectra of MeMQ in different solvents with increased polarity. **e** PL spectra of MeMQ in different solvents with increased polarity. **f** The relative PL intensity (I/I_{min}) of MeMQ in different solvents versus the normalized Reichardt's parameter (E_T^N). **g** the adiabatic energy levels of the lowest-lying (n,π^*) and (π,π^*) states in different solvents and hole-electron analysis.

(2) On the other hand, the ^{13}C NMR technique was also utilized to determine the structure of MQ in different solvents since the chemical shifts of carbon atoms of alcohol and ketone are significantly different (*J. Fluor. Chem.* 2006, 127, 780; *Dyes Pigm.* 1990, 14, 89). As shown in Supplementary Figure 24, all chemical shifts of carbon atoms are similar and lower than 160 ppm in both DMSO- d_6 and THF- d_8 , which corresponds to the keto form of MQ. And no new enol form of MQ is observed.

Supplementary Figure 24. Comparison of ^{13}C NMR spectra of MQ in DMSO- d_6 and THF- d_8 .

Therefore, it is believed that the observed changes in the emission profile for these four compounds are not affected by tautomerism. To explicitly rule out the above possibility, we have added these two figures to the Supplementary Information (Supplementary Figure 23 and Supplementary Figure 24). Besides, we also added a corresponding description in the manuscript.

“In addition, although the tautomeric balance and monomer-dimer equilibrium of pyridines were well known, the same photophysical properties of methyl-substituted MQ (MeMQ) and unchanged ^{13}C NMR spectra of MQ in different solvents strongly ruled out influence from these possibilities (Supplementary Figs. 23 and 24).⁵⁹⁻⁶² Therefore, the above results proved that PL properties in the first stage are mainly determined by the polarity and PE.”

2. Similarly, it well known that pyridones can undergo dimerization through self-complementary hydrogen bonding. Changes in monomer–dimer equilibria would be another, alternative explanation for the observed changes in emission in the presence of different

solvents which would have to be excluded before invoking the proximity effect. This could be achieved by repeating measurements at a series of different concentrations of the luminogen as the monomer–dimer equilibrium will be concentration dependent.

Response: Thanks a lot for your comments and valuable suggestion. It is acknowledged that pyridones can undergo dimerization through self-complementary hydrogen bonding as reported by many works. For four compounds (MQ, SQ, MeOSQ, and DPASQ) studied in this work, their crystal structures all displayed dimerization through self-complementary hydrogen bonding (Supplementary Figures 16-19). Nevertheless, there are two reasons and experiments that rule out the possibility that monomer-dimer equilibrium causes the observed changes in the emission of their dilute solutions with different solvents.

(1) According to previous work, the proportion of dimer increases with the increased concentration of a solution (*J. Org. Chem.* 1980, 45, 1354; *J. Am. Chem. Soc.* 1991, 113, 721; *J. Am. Chem. Soc.* 1966, 88, 1621). However, for many pyridones and their derivatives, their presence in the form of dimers is obvious only when the solution concentration is more than 10^{-3} M (2-pyridone even maintains monomer at a concentration of 4×10^{-3} M in dioxane) (*J. Am. Chem. Soc.* 1965, 87, 892). In this work, we measured their PL spectra in THF/water mixtures (Figs. 1 and 2) and different solvents (Fig. 3) at a concentration of 10^{-5} M. Thus, the proportion of dimer is too small to influence their emission properties. In addition, according to the suggestion from the review, we also measured their PL spectra in THF at a series of different concentrations from 10^{-6} to 10^{-2} M. As shown in Figure R1, four luminogens exhibit similar phenomena in that the emission intensity first increases (from 10^{-6} to 10^{-4} M) and further decreases (from 10^{-4} to 10^{-2} M) along with the increased concentration of the solution. The former is induced by the increased number of luminogens and the latter may be caused by the unstable dimer structures (based on hydrogen bonding or π - π stacking). Besides, the emission wavelength also keeps the same in different concentrations without obvious redshift, which is different from the observed redshift as shown in Fig 3. Thus, the above results clearly indicate that the proximity effect should be the reason for the observed changes in the emission of luminogens in different solvents instead of the monomer-dimer equilibrium.

Figure R1. The photoluminescence spectra of (a) MQ, (b) SQ, (c) MeOSQ, and (d) DPASQ in THF solutions with different concentrations.

(2) To exclude the formation of dimers via hydrogen bonding, the methyl-substituted MQ, namely MeMQ, was synthesized. Without the hydrogen atom, no monomer-dimer equilibrium would occur in different solvents. As shown in Supplementary Figure 23, MeMQ exhibits almost the same photophysical properties in THF/water mixtures and different solvents with different polarities. Besides, it also displays similar energy levels and hole-electron distributions. Therefore, this evidence strongly confirms that the observed changes in the emission of luminogens in different solvents are influenced by the proximity effect instead of the monomer-dimer equilibrium.

Supplementary Figure 23. The photophysical properties of methyl-substituted MQ (MeMQ). **a** The normalized absorption spectra in different solvents. Concentration = 10^{-5} M. **b** Photoluminescence (PL) spectra of MeMQ in THF/water mixtures with different water fractions (f_w). Concentration (c) = 10^{-5} M, excitation wavelength (λ_{ex}) = 320 nm. **c** Plots of relative PL intensity (I/I_0) and maximum emission wavelength versus f_w . I_0 = PL intensity at $f_w = 0\%$. $c = 10^{-5}$ M, $\lambda_{\text{ex}} = 320$ nm. **d** The normalized PL spectra of MeMQ in different solvents with increased polarity. **e** PL spectra of MeMQ in different solvents with increased polarity. **f** The relative PL intensity (I/I_{\min}) of MeMQ in different solvents versus the normalized Reichardt's parameter (E_T^N). **g** the adiabatic energy levels of the lowest-lying (n, π^*) and (π, π^*) states in different solvents and hole-electron analysis.

Thus, we also added a corresponding description in the manuscript, “In addition, although the tautomeric balance and monomer-dimer equilibrium of pyridines were well known, the same photophysical properties of methyl-substituted MQ (MeMQ) and unchanged ^{13}C NMR spectra of MQ in different solvents strongly ruled out influence from these possibilities (Supplementary Figs. 23 and 24).⁵⁹⁻⁶² Therefore, the above results proved that PL properties in the first stage are mainly determined by the polarity and PE.”

3. In the abstract, it would help to clarify that the proximity effect deals with states that are close ‘in energy’. As written currently, it could imply that the states are close in space.

Response: Thanks for your comments. According to the suggestion, we have revised the corresponding description in the abstract that “Proximity effect, which refers to the low-lying (n,π^*) and (π,π^*) states with close energy levels, usually plays a negative role in the luminescent behaviors of heterocyclic luminogens.”

4. No yields are given for the synthetic procedures in the SI. The yields should be included.

Response: Thanks a lot for the suggestion. We have added the synthesis yields of SQ, MeOSQ, and DPASQ in the synthetic procedures in the Supplementary Information.

“The pure product of SQ was 1.36 g with a yield of 55%.”

“The pure product of MeOSQ was 1.17 g with a yield of 42%.”

“The pure product of DPASQ was 1.57 g with a yield of 38%.”

5. Around lines 96-100, the authors mention that RIM effect influences the emission of MQ. What motions are being proposed as responsible for the nonradiative decay of MQ? It lacks the usual, rotatable aryl groups that are found commonly in AIE luminogens.

Response: Thanks a lot for your comments. For luminogens with AIE effect, both rotations of rotatable units and vibration of the molecular skeleton can affect their emission (*Chem. Res. Chin. Univ.* 2021, 37, 1; *Natl. Sci. Rev.* 2020, 8, nwa260; *Chem. Commun.* 2019, 55, 1879; *Angew. Chem. Int. Ed.* 2021, 60, 22417). For MQ, we have studied its motion modes in both THF solution and crystalline state using theoretical calculations and reorganization energy analysis, which can reflect the intrinsic geometry change of molecules between the ground state and excited state (*Mater. Chem. Front.* 2019, 3, 1143; *Natl. Sci. Rev.* 2017, 4, 224). As shown in Figure R2, the total reorganization energy decreases from 4557 cm^{-1} in the isolated state to 3328 cm^{-1} in the crystalline state, suggesting the mechanism of restriction of intramolecular motions. Especially, two typical motion modes that are active in the isolated state but limited in the crystalline state are selected (Figure R2b). This first one at the wavenumber of 149.81 cm^{-1} corresponds to the rotation of the methyl group and the other one at 470.83 cm^{-1} corresponds to the stretching and vibration of the MQ skeleton. Therefore, the above results suggest that both rotation and vibration can influence the emission of MQ.

Figure R2. Plots of reorganization energy vs. normal mode wavenumber of MQ in the (a) isolated and (c) crystalline states. Inset: proportions of bond length, bond angle, and dihedral

angle contributed to total reorganization energy. **(b)** Two typical motion modes and displacement vectors of MQ in the isolated state.

6. Information should be given about the red lines added to the data in Figures 3e-h. How have they been fitted? What are the chi squared values? It may be better not to show any line if there is not a clear/unambiguous trend, e.g., especially for panel g.

Response: Thanks a lot for your valuable suggestions. The points shown in Fig. 3e-h represent the relative PL intensity of compounds in different solvents, and the red lines are linear fittings for these separated points. We hope to use such a line to show the change in fluorescence intensity of these four compounds in different solvents with increased polarity. According to your suggestion, we have added the R-squared values to the figure. Due to the trend for MeOSQ (Fig. 3g) is not clear (R squared value for linear fitting is 0.125), the red line is deleted. Besides, we have added the information for the red lines in the caption of Fig. 3.

“**e-h** The relative PL intensity (I/I_{\min}) of **(e)** MQ, **(f)** SQ, **(g)** MeOSQ, and **(h)** DPASQ in different solvents versus the normalized Reichardt’s parameter (E_T^N). I_{\min} = the lowest PL intensity of the target compound in different solvents. The linear fitting (red line) represents the trend of I/I_{\min} in different solvents with increased polarity, R^2 = goodness of fit”

Fig. 3 Photoluminescence (PL) properties of MQ, SQ, MeOSQ, and DPASQ in different solvents. **a-d** Normalized PL spectra of **(a)** MQ, **(b)** SQ, **(c)** MeOSQ, and **(d)** DPASQ in different solvents with increased polarity (*n*-hexane (hexane) < toluene < dioxane < tetrahydrofuran (THF) < dichloromethane (DCM) < dimethyl sulfoxide (DMSO) < acetonitrile (ACN) < methanol (MeOH)). Concentration = 10^{-5} M. The arrow indicates the increase of solvent polarity. **e-h** The relative PL intensity (I/I_{\min}) of **(e)** MQ, **(f)** SQ, **(g)** MeOSQ, and **(h)** DPASQ in different solvents versus the normalized Reichardt’s parameter (E_T^N). I_{\min} = the lowest PL intensity of the target compound in different solvents. The linear fitting (red line) represents the trend of I/I_{\min} in different solvents with increased polarity, R^2 = goodness of fit.

7. Is there an explanation that can be given as to why the π, π^* states are stabilised by increased

solvent polarity while the n,π^* states are destabilised?

Response: Thanks a lot for your comments. The energy level of (π,π^*) and (n,π^*) states are closely related to their stabilization in different solvents, as shown in Figure R3. For (π,π^*) state, the molecular polarity in the excited state is always larger than that in the ground state. From nonpolar solvent to polar solvent, both π and π^* states are stabilized. However, the strength of energy reduction in the excited state is greater than that in the ground state, so the energy required to achieve the (π,π^*) state becomes smaller, resulting in the decreased energy level of (π,π^*) state. On the contrary, the lone pair of electrons is strongly stabilized in polar solvents, and the ground-state energy level of n has a larger drop than the excited-state energy level of π^* . Overall, the energy required to realize the (n,π^*) state increases accordingly. Based on the above mechanism, it is reasonable that the energy level of the (π,π^*) state gradually decreases while that of the (n,π^*) state gradually increases with the increase of solvent polarity as shown in Fig. 4.

Figure R3. The energy levels of (n,π^*) and (π,π^*) states in nonpolar and polar solvents.

8. Is there a reason that the authors have chosen the PBE0-D3/6-31G(d,p) method for TD-DFT? Has it been benchmarked to the experimental data, or have the authors screened other functionals and basis sets? Coupled cluster or ADC methods would likely give more reliable results for the excited state energy levels and are feasible for the small molecules being investigated. This point is particularly important given that calculated energies are a key component of the proposed ‘induced’ and ‘suppressed’ proximity effects.

Response: Thanks a lot for your comments. In general, the theoretical level of PBE0-D3/6-31G(d,p) shows advantages in terms of calculation time and accuracy for TD-DFT calculations. There are several reasons for us to utilize this theoretical level in this work: (1) PBE0 is a widely utilized method for TD-DFT calculation of molecules with the locally excited feature, which yields a mean signed error of *ca.* 0.05 eV and mean absolute error of *ca.* 0.25 eV benchmarked by some theoretical chemist (*Phys. Chem. Chem. Phys.* 2011, 13, 16987; *Chem. Soc. Rev.* 2007, 36, 1724). To propose the ‘induced’ and ‘suppressed’, the core compounds of MQ and SQ mainly show locally excited features, so the theoretical method is suitable for this system. (2) This method of PBE0-D3/6-31G(d,p) has been widely accepted and utilized to calculate particularly important energy levels of singlet/triplet states to evaluate luminescent performance in many high-impact journals (e.g., *Sci. Adv.* 2021, 7, eabj2504; *Angew. Chem. Int. Ed.* 2023, 62, e202214769; *Matter* 2023, 6, 1231; *Nat. Commun.* 2015, 6, 8476; *J. Am. Chem. Soc.* 2023, 145, 3, 1945; *Angew. Chem. Int. Ed.* 2015, 54, 15231). (3) It is acknowledged that coupled cluster or ADC methods could provide high-precision results for energy levels. However, ADC method is not supported by Gaussian. And the popular coupled cluster method (e.g., CCSD or CCSD(T)) is usually utilized for single-point calculations of small molecules, which is not feasible for geometry optimization of molecules with more than 30 atoms. Besides,

both methods consume a lot of computing power and time. (4) Taking MQ as an example, we also compared the experimental and theoretical values of emission wavelength using different functionals (Table R1). Among different functionals, the wavelength calculated by PBE0-D3 was the closest one to the experimental value.

Table R1. The experimental and theoretical values of the emission wavelength of MQ in DCM solution using different functionals and basis set of 6-31G(d,p).

-	experimental value	theoretical values			
		PBE0-D3	M062X-D3	CAM-B3LYP-D3	ω B97XD
Wavelength	399 nm	365 nm	351 nm	354 nm	355 nm

Therefore, the above discussion indicates that PBE0-D3/6-31G(d,p) is a feasible method for TD-DFT of these compounds in this work, and it could provide particularly important energy levels with reasonable errors and computing time for the proposed ‘induced’ and ‘suppressed’ PE.

Accordingly, we added the reasons and references for this calculation method in the method part of computational details that “The geometries of all compounds in the ground state (S_0) were optimized using the density functional theory (DFT) method at the PBE0/6-31G(d,p) level with Grimme’s DFT-D3 correction, which was widely utilized to evaluate luminescent materials and their energy levels with comparatively high accuracy.⁷¹⁻⁷⁴ The excited-state geometries were optimized using the time-dependent DFT method at the same level of theory.”

9. On line 223, the authors imply that the convergence of the π, π^* and n, π^* energy levels causes a redshift in the emission. It’s not immediately obvious why that would be. Can more explanation be given? Is it not the case that the redshift is caused by the greater solvent dipole relaxation in more polar solvents, independent of the spacing of these two energy levels relative to one another?

Response: Thanks a lot for your comments. With the increased polarity of solvents, the energy level of the (n, π^*) state increases, and the energy level of the (π, π^*) state decreases, respectively. As a result, this decreased (π, π^*) state results in the redshifted emission of MQ. Meanwhile, the energy gap between these two states becomes smaller, which gradually enhances the vibronic coupling of (n, π^*) and (π, π^*) states and PE impact. Finally, this change boosts the radiative decay and fluorescence from the (π, π^*) state. In other words, the redshift of emission is only affected by the (π, π^*) energy level, while the increased PL is controlled by the convergence of (n, π^*) and (π, π^*) states. To reduce misunderstandings, we have revised the corresponding description in the manuscript.

“Meanwhile, the gradually enhanced vibronic coupling of (n, π^*) and (π, π^*) states induced the PE and boosted the radiative decay from (π, π^*) state, which was the reason why MQ showed PL intensity enhancement. The decreased energy level of (π, π^*) state also resulted in the redshift of the enhanced emission.”

10. The amount of TFA in Figure 6 would be better given as a concentration rather than a volume. The volume of the DCM solution of the luminogen is not given, so there is no way for the reader to know the relative stoichiometry/concentration of the TFA relative to the luminogen.

Stating the concentration of TFA avoids this problem and is the most informative way to report it for the reader.

Response: Thanks a lot for your valuable suggestions. We have remeasured the responsive properties of these three compounds to TFA with different concentrations, the results are the same as the previous one. Therefore, we have revised Fig. 6a-c and Supplementary Figure 26 and indicated the concentration of TFA added to the solutions.

Fig. 6 Dynamic responses to acid/based stimuli and applications. a-c Top panel: photoluminescence spectra of (a) MQ, (b) MeOSQ, and (c) DPASQ solutions in dichloromethane upon adding TFA. The concentration of luminogens is 10^{-4} M. Bottom panel: fluorescent photographs of responses towards protonation by addition of TFA to solutions or fuming films deposited on filter paper by TFA gas. **d** Fluorescent intensity of protonated and deprotonated DPASQ films upon sequential NH_3 and TFA fuming. **e** Photographs of fluorescent responses of MQ and DPASQ upon NH_3 and TFA exposure for sensor and information encryption.

Supplementary Figure 26. Absorption spectra of (a) MQ, (b) MeOSQ, and (c) DPASQ solutions in DCM (2 mL) upon the addition of trifluoroacetic acid (TFA). The concentration of luminogens is 10^{-4} M.

11. The absorbance data in SI Figure 19 does not appear to show an isospeptic point for the titration of TFA into the solutions of MQ, MeOSQ and DPASQ, which is not consistent with a simple equilibrium from unprotonated to protonated luminogen. It may be more consistent with the presence of tautomers and/or dimers. This data should be considered in response to points 1 and 2.

Response: Thanks a lot for your comments. As discussed in response to points 1 and 2, the possibility of tautomerism and monomer-dimer equilibrium affecting the changes in the emission of luminogens in different solvents is ruled out. To further support our conclusion, we also measured the response of MeMQ, methyl-substituted MQ, to trifluoroacetic acid (TFA) in the DCM solution. As shown in Figure R4, its absorption spectra and photoluminescence spectra are almost the same as that of MQ, suggesting their same photophysical properties and electronic structure upon the addition of TFA.

Figure R4. a Absorption spectra of MeMQ in DCM (2 mL) upon the addition of trifluoroacetic acid (TFA). b Photoluminescence spectra of MeMQ in DCM (2 mL) upon the addition of TFA. Concentration = 10^{-4} M.

It is acknowledged that an isosbestic point is usually observed for many simple equilibria. However, the isosbestic point may disappear in some complex equilibrium reactions (e.g., *Inorg. Chem.* 1991, 30, 906; *J. Am. Chem. Soc.* 1981, 103, 2667; *ChemBioChem* 2023, 24,

e202200744). Actually, the absorption and fluorescence changes upon protonation of these three compounds is not a simple equilibrium, which is also accompanied by the effect of TFA on polarity and energy levels of (n,π^*) and (π,π^*) states of luminogens, especially for MQ. As a result, no isosbestic point is observed in the absorption spectra of MQ (Supplementary Figure 26). Correspondingly, the change of its photoluminescence spectra is not linear, which shows first increased and further decreased intensity of emission at 397 nm.

Response to the Comments of Reviewer 2:

In this manuscript, Tang et al. reported an electron-donor engineering strategy to manipulate the proximity effect (PE) of four methylquinoxaline (MQ) derivatives and revealed the impact of PE on AIEgens. Spectroscopic analysis was investigated to show their fluorescent change toward water addition and aggregation, and theoretical calculations systematically illustrated their electronic structures and energy-level variation of (n, π^*) and (π, π^*) states in solvent environments with different polarities. In addition, they studied the responses of these AIEgens to acid/base stimuli and demonstrated potential applications. Generally, PE has been regarded as the nonradiative decay channel for luminogens for many years, but it is interesting and significant that the authors revealed both the positive and negative impacts of PE for AIEgens with different electronic states. The present manuscript is an important contribution to AIE research as it (i) connects PE and AIE for the first time and (ii) provides a new perspective to design and regulate the photophysical properties of heterocycle-containing AIEgens. The work is innovative, well-justified, and generally well-written. Therefore, this manuscript is recommended for publication subject to the following minor issues.

Response: We really appreciate the recognition of the novelty and importance of our work. We have revised the manuscript according to these valuable suggestions from the reviewer.

1. Restriction of intramolecular motion (RIM) is proved as the key mechanism for AIE effect. However, in the introduction, the authors claimed that the combination of PE and RIM could replenish AIE mechanisms. Please explain the linkage between PE and RIM and how PE can provide new insights into the AIE mechanism.

Response: Thanks a lot for your comments. Restriction of intramolecular motion (RIM) is a fundamental mechanism for AIE phenomena, which mainly focuses on geometry change and intramolecular motions of molecules and highlighted the restriction of these motions in aggregate state or rigid matrix to block nonradiative decay and realize enhanced luminescence (*Chem. Res. Chin. Univ.* 2021, 37, 1; *Natl. Sci. Rev.* 2020, 8, nwaa260; *Adv. Sci.* 2017, 4, 1600484). On the other hand, apart from flexible geometry and motions, electronic structures bring some unique photophysical properties, especially for heteroatom-containing compounds. Proximity effect (PE) is one kind of electronic interaction that can also influence the emission properties of luminogens. For some luminogens with AIE effect and closely related (n, π^*)/(π, π^*) states, these two impacts can synergetically manipulate their photophysical properties. Therefore, the study of PE in this work can provide new insights into the AIE phenomenon and mechanism from three perspectives: (1) reveal the both positive and negative role of PE on heteroatom-containing luminogens with AIE effect; (2) understand the PE and RIM impacts on luminescent behaviors during the different stages of aggregation process; (3) provide a synergetic strategy to regulate photophysical properties of heterocyclic AIEgens through PE and RIM.

2. It's reasonable that DPASQ exhibits the twisted intramolecular charge transfer effect (TICT) toward water addition (Fig. 2), while MeOSQ was different due to its weak D-A interaction. The author also provided their PL spectra in different solvents and conducted theoretical calculations to support the TICT effect. However, I am curious about their fluorescence change and AIE curve in polar solvents, e.g., DMSO/H₂O or MeOH/ H₂O mixtures.

Response: Thanks for your review and comments. Since the methylquinoxalin group is inherently an electron acceptor and triphenylamine is a strong electron donor, DPASQ exhibits

a strong D-A interaction. This structural feature endows it with the ability to show TICT effect in polar solvents. Indeed, PL intensity of DPASQ first decreases and further increases toward water addition in THF/water mixtures (Fig. 2d and e), which is a typical TICT and AIE phenomenon (*Chem. Soc. Rev.* 2021, 50, 12656; *J. Am. Chem. Soc.* 2016, 138, 6960). As a comparison, MeOSQ with a weak donor exhibits gradually increases PL intensity in the THF/water mixtures without TICT effect (Fig. 2b and c). According to your suggestion, we also measured the fluorescence and AIE curve in MeOH/water mixtures (Supplementary Figure 20). As expected, MeOSQ shows a similar AIE curve and gradually enhanced fluorescence intensity with the addition of water into the MeOH/water mixtures. For DPASQ, it also displays first decreased and further increased PL intensity, supporting the TICT and AIE effects. Due to MeOH and water being polar solvents, the intensity decline is comparatively weaker than that in THF/water mixtures, and the intensity enhancement after aggregation ($f_w \geq 50$) is more obvious. Therefore, the fluorescent spectra and AIE curves in two kinds of mixtures together confirm the strong D-A interaction of DPASQ and its TICT effect in polar solvents.

Accordingly, we have added this figure in the Supplementary Information as Supplementary Figure 20. We also added a description of fluorescent spectra and AIE curves of DPASQ in the manuscript.

“The fluorescent spectra and PL intensity change of DPASQ in MeOH/water also supported its TICT and AIE features (Supplementary Fig. 20).”

Supplementary Figure 20. Photophysical properties of MeOSQ and DPASQ in MeOH/water mixtures. **a** Photoluminescence (PL) spectra of MeOSQ in MeOH/water mixtures with different water fractions (f_w). Concentration (c) = 10^{-5} M, excitation wavelength (λ_{ex}) = 390 nm. **b** Plots of relative PL intensity (I/I_0) versus f_w . I_0 = PL intensity at $f_w = 0\%$. **c** PL spectra of DPASQ in MeOH/water mixtures with different f_w . $c = 10^{-5}$ M, $\lambda_{ex} = 430$ nm. **d** Plots of I/I_0

versus f_w .

3. As illustrated in Fig 4, the (n,π^*) state was always lower than (π,π^*) state of MQ, resulting in the anti-Kasha's emission. However, achieving such kind of emission is a difficult issue and requires strict conditions. The red-shifted emission of MQ in THF/ H₂O seems to support this anti-Kasha's behavior. Are there other reasons why the emission from MQ came from the higher (π,π^*) state instead of the (n,π^*) state?

Response: Thanks a lot for the consideration and comments. It is true that realizing anti-Kasha's emission is difficult and may require strict conditions: (1) a large energy difference between the higher excited state (S_n , $n \geq 2$) and S_1 , which suppresses the rate of internal conversion; (2) the higher S_n and S_1 states are vibronically mixed with small energy difference (*Nat. Commun.* 2017, 8, 416; *Angew. Chem. Int. Ed.* 2013, 52, 13449; *J. Lumin.* 2004, 109, 221). Accordingly, the (π,π^*) and (n,π^*) states of MQ with small energy difference matched the above requirement for anti-Kasha's emission (Fig. 4). Besides, the gradually decreased gap between these two states along with the increased solvent polarity would enhance vibronic coupling, which was consistent with its enhanced intensity of anti-Kasha's emission.

On the other hand, there are two more reasons that may further support the emission from the (π,π^*) state: (1) the (n,π^*) state with small oscillator strength (f) is usually regarded as a nonradiative decay channel. However, the (π,π^*) state with a large value of f is considered an emissive state. (2) As shown in Fig. 4, the energy level of (π,π^*) state gradually decreases with the increased solvent polarity, which is consistent with the redshifted emission of MQ in different solvents with increased polarity. In contrast, the energy level of the (n,π^*) state gradually increases with the increased solvent polarity. As a result, it is believed that the emission from MQ originates from the higher (π,π^*) state with anti-Kasha's feature.

4. The first stage behaviors of these four compounds toward water addition are carefully shown in Fig. 5. On the other hand, a simple description of their PL change after aggregation should be added as a complete illustration, although they show similar behavior.

Response: Thanks a lot for your comment and good suggestion. We have added a general description for their PL change after aggregation in the second stage of adding water to the THF solution, which should be a complete illustration of their photophysical properties (Fig. 5).

“Nevertheless, in the second stage of adding water to the THF solution ($f_w \geq 60$ for MQ and SQ, and $f_w \geq 70$ for MeOSQ and DPASQ), four AIEgens displayed similar AIE phenomenon with gradually enhanced PL intensity after aggregation.”

5. Please explain why MeOSQ and DPASQ theoretically exhibit a slight enhancement of PL intensity and red-shifted emission with increased solvent polarity, as described on page 9.

Response: Thanks for your comment. Compared to MQ and SQ, the energy levels of the (π,π^*) state are always lower than the (n,π^*) state in MeOSQ and DPASQ, and the energy gap between them is comparatively large (Fig. 4). Therefore, the proximity effect is very weak in MeOSQ and DPASQ, which can not quench their excitons. Theoretically, the increased energy gap between (π,π^*) and (n,π^*) states shows a very weak influence to further enhance emission from the emissive (π,π^*) state, resulting in a slight enhancement of PL intensity. On the other hand,

as shown in Fig. 4, it is acknowledged that the energy level of (π, π^*) state gradually decreases with the increased solvent polarity, which should produce red-shifted emission. Besides, the emission wavelength and intensity are also influenced by the important charge transfer or twisted intramolecular charge transfer process, which was discussed in the manuscript.

Therefore, to better describe their intrinsic photophysical properties, we have revised the above statement that “Due to the negligible PE on them, theoretically, these two compounds should exhibit a slight enhancement of PL intensity with the increased solvent polarity. However, apart from PE, it needs to be emphasized that TICT is also a dominant effect in strong D–A systems with a rotatable bond, which always induces decreased PL intensity and redshifted λ_{em} with the increase of solvent polarity.”

6. Please list Reichardt’s parameters and normalized parameters for each solvent used in Fig. 3.

Response: Thanks for your suggestion. Reichardt’s parameters and normalized Reichardt’s parameters are scales of solvent polarity, which are obtained from a widely cited review (*Chem. Rev.* 1994, 94, 2319). According to the suggestion, we have listed Reichardt’s parameters and normalized parameters for each solvent, as well as the relative photoluminescence intensity (I/I_{min}) of MQ, SQ, MeOSQ, and DPASQ in different solvents in the Supplementary Information. This corresponding reference was also cited.

Supplementary Table 2. Reichardt’s parameters and normalized Reichardt’s parameters of different solvents, and relative PL intensity (I/I_{min}) of MQ, SQ, MeOSQ, and DPASQ in different solvents.^[2]

	n -Hexane	Toluene	Dioxane	THF	DCM	DMSO	ACN	MeOH
Reichardt’s parameter (E_T)	31.0	33.9	36.0	37.4	40.7	45.1	45.6	55.4
Normalized Reichardt’s parameter (E_T^N)	0.009	0.099	0.164	0.207	0.309	0.444	0.460	0.762
Relative PL intensity								
MQ	1.000	1.618	1.434	1.268	2.460	1.902	2.475	5.461
SQ	1.000	1.312	1.297	1.163	1.182	1.715	1.606	1.892
MeOSQ	1.123	2.191	2.191	1.895	1.867	2.191	1.000	1.139
DPASQ	13.221	16.971	23.045	23.407	17.578	5.324	5.165	1.000

7. Regarding the Methods part, the authors claimed that “all chemicals were utilized as received without purification”. However, “MQ was purified with column chromatography”, as stated in SI. Please check and unify the description.

Response: Thanks a lot for your careful review. Since 3-methylquinoxalin-2(1H)-one (MQ) was utilized for photophysical measurements and comparison with other compounds, the purchased MQ was first purified with column chromatography to ensure its purity. Therefore, we have revised and unified the relevant description in the Methods part of the manuscript and Supplementary Information.

“MQ was purified using column chromatography with dichloromethane, and other chemicals were utilized as received without further purification.”

“The purchased 3-methylquinoxalin-2(1H)-one (MQ) which appeared to be black powders was purified using column chromatography with dichloromethane followed by recrystallization to obtain pale-yellow crystals.”

8. There are several typos:

(i) Page 2, “aggregation-induced emission (AIE), ..., which referred to the luminogens”. AIE should refer to an effect or phenomenon instead of luminogens.

(ii) Page 5, “typical” should be “typically”.

Response: Thanks for your comments. We have revised the above typos and carefully checked the full manuscript.

“Fortunately, aggregation-induced emission (AIE), an opposite effect to ACQ, was coined, which referred to the phenomenon that luminogens were strongly emissive in the aggregate state but nonemissive in their dilute solutions.”

“Typically, the absorption spectra of DPASQ showed an obvious solvent effect without fine structures, indicating its strong charge transfer (CT) feature.”

Response to the Comments of Reviewer 3:

Using four kinds of AIE compounds, the authors claim that the enhancement of luminescence intensity at the initial stage of aggregate formation can be controlled by solvent polarity based on the PE effect. They finally state that this mechanism can be applied to AIE in heterocycles. Although it is interesting as a proposal on the AIE mechanism, it is considered that there are few experimental examples to claim generality. Therefore, I judged that this manuscript is not suitable for publication at this stage. The following points must be considered.

Response: We appreciate the assessment and comments from the reviewer. We have revised the manuscript according to your valuable suggestions. We hope the revised manuscript and response could address all concerns.

1. Emission mechanism of SQ is similar to that of pyrene aldehyde, while the others show emission from the CT states. It is an exaggeration to compare those with different emission mechanisms under the same theoretical framework, even though they all show AIE.

Response: Thanks a lot for your comments and consideration. Actually, in this work, we hope to reveal a detailed mechanism and structure-property relationship for heterocyclic AIEgens with close energy levels of (π,π^*) and (n,π^*) states and different electron-donating groups, instead of using an exaggerated theory to compare luminogens with totally different mechanisms. And the main feature of this work is realizing the regulation of PE on the photophysical properties of these heterocyclic luminogens through an electron-donor strategy.

There are two main reasons to support our interpretation. (1) As reported by some previous works (e.g., *Tetrahedron* 2012, 68, 6177; *Chem. Eur. J.* 2013, 19, 9760; *J. Org. Chem.* 2012, 77, 8, 3986), pyrene aldehyde (PA) shows typical solvatochromism and comparatively high quantum yield in polar solvents, which is influenced by intersystem crossing and proximity effect (PE). In terms of energy levels, PA displayed closely related (π,π^*) and (n,π^*) states, and the energy level of (n,π^*) is slightly higher than (π,π^*) state. Therefore, SQ and PA are similar in terms of photophysical properties and electronic structures. For MQ, it also shows similar photoluminescence properties as SQ with gradually enhanced intensity from apolar to polar solvents. However, the energy level of (n,π^*) state is lower than the (π,π^*) state in MQ. Meanwhile, the energy level of its (π,π^*) state is higher than that of the (n,π^*) state in MeOSQ and DPASQ, and the gap between these two states of DPASQ is bigger than MeOSQ. Therefore, these four compounds with gradually changed energy levels from MQ to SQ, MeOSQ, and DPASQ provide a good platform to investigate the regulation of PE and how PE influences their photophysical properties. (2) As a parent core, MQ is an intrinsic electron acceptor and shows gradually enhanced PE with increased solvent polarity. With the addition of an electron-donating phenyl group, SQ shows gradually declined PE but still weak CT features. Thus, PE mainly controls its luminescent properties. For

MeOSQ with a stronger donor, the decreased PE and enhanced CT compete with each other, resulting in a net effect of very weak intensity enhancement. For DPASQ with the strongest electron donor, the TICT effect is obvious and overwhelms PE, displaying TICT-controlled luminescent properties (Figs. 1 and 2). Although a new mechanism of TICT is introduced when PE becomes weak, the key results of this work are to illustrate (i) the positive and negative impacts of PE for heterocyclic luminogens with close (π, π^*) and (n, π^*) states in energy; (ii) the competitive relationship and synergetic effect of PE and TICT for the photophysical property of these luminogens; and (iii) the strength of PE could be regulated by electron-donating ability; As discussed above, we have no intention of summarizing these AIEgens with different mechanisms under the same theoretical framework.

To avoid misunderstanding, we have revised the corresponding description in the manuscript that “(4) When the electron-donating ability of the π -conjugated donor was further strengthened, the TICT effect was obvious, and the PE disappeared due to the very large energy gap. Accordingly, TICT became the key mechanism to influence photophysical properties, which endowed DPASQ with gradually decreased PL intensity and dramatically redshifted λ_{em} toward water addition (Fig. 5e).”

Besides, we also addressed the similar phenomenon of pyrene aldehyde when discussing the emission of MQ and SQ that “MQ displayed monotonically increased PL intensity with a maximum I/I_{min} of 5, and SQ showed a similar enhancement with a smaller slope (the maximum $I/I_{min} = 1.9$). A similar phenomenon was also observed in some pyrene aldehyde systems.⁵⁶⁻⁵⁸”

2. Also, to insist generality of this effect in all heterocyclic systems, further examples are needed. At least, with pyrimidines which have different aza-substituted positions and heteroles which have different sizes, the authors need to show the similar effects.

Response: Thanks a lot for your review and concerns. As discussed in the manuscript, proximity effect occurs when the lowest (π, π^*) and (n, π^*) states are close in energy. If (n, π^*) state is lower than (π, π^*) state, the increased polarity of solvents can decrease the energy gap between them and increase PE, resulting in the induction of proximity effect. In contrast, if (π, π^*) state is lower than (n, π^*) state, the increased polarity of solvents would further increase their energy gap and limit PE, producing the suppression of proximity effect. Thus, the close energy levels of the lowest (π, π^*) and (n, π^*) states are prerequisites, which means that not all heterocyclic systems or pyrimidines can exhibit similar effects observed in this work. Besides, the impact of proximity effect in some heterocyclic systems has been reported, such as pyrene aldehyde (*Tetrahedron* 2012, 68, 6177), pyrazine (*J. Chem. Phys.* 1976, 65, 1219), 6(5H)-phenanthridinone (*J. Phys. Chem. A* 2007, 111, 193), 5,6-trimethylenecytosine and 5,6-trimethylenouracil (*Phys. Chem. Chem. Phys.* 2007, 9, 3206), 9H-adenine (*J. Am. Chem. Soc.* 2005, 127, 6257; *Proc. Natl. Acad. Sci. U.S.A.* 2006, 103, 8691), etc. These works already verified the generality of proximity effect on excited-state decay and photophysical properties of heterocyclic systems with close (π, π^*) and (n, π^*) states

in energy. In fact, this work is the first time to build the bridge between the AIE community and proximity effect. The main focus is to illustrate the impact of proximity effect on the photophysical properties of AIEgens and realize the regulation of proximity effect through an electron-donor strategy based on a skeleton of methylquinoxaline (MQ) with intrinsic proximity effect.

On the other hand, according to the suggestion from the reviewer, we selected three other examples (acridone, 7-methoxycoumarin, and pyrene aldehyde) with different molecular skeletons and measured their photoluminescence spectra in different solvents (Figure R5). The results show a general trend that the emission intensity gradually increases and the emission wavelength gradually redshifts along with the increased polarity of solvents. This effect is similar to these four compounds in this work (Fig. 3). In addition, we also calculate their energy levels as shown in Figure R6. The result shows their close energy levels of (n,π^*) and (π,π^*) states, and the energy level of (n,π^*) state gradually increases while that of (π,π^*) state gradually decreases with the increased solvent polarity. Interestingly, acridone even shows energy-level crossing with the lowest state changed from the (n,π^*) state in hexane to (π,π^*) state in methanol, which may explain its high sensitivity to solvent polarity with $I/I_{\min} = 390$ (Figure R5). Thus, the above results support our proposed proximity effect and its impact on the photophysical properties of some heterocyclic systems with close energy levels of the lowest (π,π^*) and (n,π^*) states.

Figure R5. a-c Photoluminescence (PL) spectra of (a) acridone, (b) 7-methoxycoumarin, and (c) pyrene aldehyde in different solvents with increased polarity. e-f The relative PL intensity (I/I_{\min}) of (d) acridone, (e) 7-methoxycoumarin, and (f) pyrene aldehyde in different solvents versus the normalized Reichardt's parameter (E_T^N). I_{\min} = the lowest PL intensity of the target compound in different solvents, $R^2 =$

goodness of fit.

Figure R6. The adiabatic energy levels of the lowest-lying (n,π^*) and (π,π^*) states in different solvents and hole-electron analysis of (a) acridone, (b) 7-methoxycoumarin, and (c) pyrene aldehyde.

Accordingly, to clarify the scope of PE, we have revised the relevant description in the manuscript.

“The above mechanistic perspective suggests the successful manipulation of PE and photophysical properties for heterocycle-containing AIE systems with close energy levels of (n,π^*) and (π,π^*) states.”

“PE plays an essential role in controlling the photophysical properties of heterocycle-containing compounds with close energy levels of the lowest (π,π^*) and (n,π^*) states, which is usually regarded as a nonradiative decay channel for luminescent materials.”

3. To discuss the solid-state luminescence properties in AIE behaviors, the morphology must be clarified. In particular, since molecules can aggregate or crystallize at the initial stage by increasing water contents in the aggregation experiments, comparisons between undefined structures can misinterpret results if morphology is not unified.

Response: Thanks a lot for your careful consideration and suggestions. For solid-state

samples, we have measured their powder X-ray diffraction patterns and added this figure as Supplementary Figure 15c. The results show that all of them are in the crystalline state instead of the amorphous state.

Supplementary Figure 15. (a) Photoluminescence spectra, (b) fluorescent photographs and absolute quantum yield (QY), and (c) powder X-ray diffraction patterns of MQ, SQ, MeOSQ, and DPASQ in the crystalline state.

Accordingly, we have clarified their crystalline state in the manuscript when discussing the solid-state luminescence properties.

“The absolute quantum yield (Φ) of MQ in the crystalline state (2.1%) was higher than the pure solution (0.1%) due to multiple hydrogen-bonding interactions and RIM effect”

“In addition, its crystalline sample with close packing showed a bright yellow light with a Φ of 10.2%, which was higher than the other three compounds (Supplementary Figs. 15 and 19)”

“In the crystalline state, they displayed varying Φ (from 2.1% to 10.2%) but the same nature of fluorescence as suggested by their nanosecond lifetimes (Supplementary Fig. 21).”

In addition, the state of molecules (isolated or aggregate state) at the initial stage by increasing water contents in the aggregation experiments was monitored using the dynamic light scattering (DLS) technique. DLS is a typical method in physics to determine the formation and particle size of aggregates, which is widely utilized in biological, polymer, and AIE fields (*Analyst*, 2013, 138, 1212; *Adv. Energy Mater.* 2023, 13, 2203009; *Adv. Mater.* 2015, 27, 5158; *Angew. Chem. Int. Ed.* 2020, 59, 10327). Therefore, as shown in Supplementary Figure 14, nanoparticles were only detected when water fractions ≥ 60 or 70%. The results suggested that molecules of these four compounds kept the isolated state with aggregation at the initial stage by adding water into the THF/water mixtures. Therefore, the change of their PL spectra at the initial stage is not affected by aggregation.

Supplementary Figure 14. The size distribution of (a) MQ, (b) SQ, (c) MeOSQ, and (d) DPASQ in the THF/water mixtures with different water fractions. Concentration = 10^{-5} M.

4. What about the triplet state properties of your molecules? In general, the low-lying (n,π^*) state of carbonyl compounds is responsible for their fast intersystem crossing.

Response: Thanks a lot for your consideration. To answer this concern, we have measured the lifetimes of these four compounds at room temperature (Supplementary Fig. 21). The result suggests that their emission belongs to fluorescence with nanosecond lifetimes, and no phosphorescence is observed although they showed low-lying (n,π^*) state.

It is acknowledged that the low-lying (n,π^*) state of the carbonyl group is beneficial for room-temperature phosphorescence (RTP) (*Nat. Rev. Mater.* 2020, 5, 869; *Chem* 2016, 1, 592; *J. Am. Chem. Soc.* 2019, 141, 1010). However, the feature alone can not guarantee the occurrence of RTP, which usually requires several conditions including

(1) strong spin-orbital coupling, (2) small energy gaps between excited singlet and triplet states, (3) fast phosphorescent decay from the T_1 state, (4) slow nonradiative decay and quenching rate, and so on. Especially, according to the El-Sayed rule, the rate of intersystem crossing is dependent on features of both singlet and triplet states, and only the ISC from $^1(\pi,\pi^*)$ to $^3(n,\pi^*)$ or from $^1(n,\pi^*)$ to $^3(\pi,\pi^*)$ is favored. Therefore, not all compounds containing carbonyl groups have the characteristics of RTP, and so do the four compounds in this work. In contrast, the low-lying (n,π^*) state may be responsible for the fast intersystem crossing and nonradiative decay from the triplet state, which is consistent with the lowest quantum yield of MQ possessing a low-lying (n,π^*) state.

Accordingly, to better clarify the nature of their luminescence, we have added the results of lifetime measurements in the Supplementary Information as Supplementary Figure 21 and a description of it in the manuscript that “In the crystalline state, they displayed varying Φ (from 2.1% to 10.2%) but the same nature of fluorescence as suggested by their nanosecond lifetimes (Supplementary Fig. 21).”

Supplementary Figure 21 Time-resolved photoluminescence decay curves of (a) MQ, (b) SQ, (c) MeOSQ, and (d) DPASQ taken at their corresponding maximum emission wavelength (λ_{em}) in the crystalline state.

5. In Figures 1b, 1d, 2b, 2d and 3, it is difficult to recognize the relationships between the peak shift and parameter changes. Please insert the arrow and rearrange line colors, such as rainbow color, to readily understand the increase of water contents or polarity.

Response: Thanks a lot for your comments. According to your valuable suggestion, we have revised Fig. 1 and Fig. 2 by inserting an arrow and rearranging these lines with rainbow colors to indicate the change in the direction of PL spectra with the increased water fractions in THF/water mixtures. Besides, we have inserted the arrow and rearranged multiple lines with rainbow colors in Fig. 3 to show the increase of solvent polarity. We hope our revision can help readily understand the change of fluorescence in these figures.

Fig. 1 Photophysical properties of MQ and SQ. **a** The chemical structures of MQ and SQ. **b** Photoluminescence (PL) spectra of MQ in THF/water mixtures with different water fractions (f_w). Concentration (c) = 10^{-5} M, excitation wavelength (λ_{ex}) = 320 nm. **c** Plots of relative PL intensity (I/I_0) and maximum emission wavelength versus f_w . I_0 = PL intensity at $f_w = 0\%$. Inset: fluorescent photographs of MQ in the THF/water mixtures with f_w of 0% and 90%, respectively. **d** PL spectra of SQ in THF/water mixtures with different f_w . $c = 10^{-5}$ M, $\lambda_{ex} = 350$ nm. **e** Plots of I/I_0 and maximum emission wavelength versus f_w . Inset: fluorescent photographs of SQ in the THF/water mixtures with f_w of 0% and 90%, respectively.

Fig. 2 Photophysical properties of MeOSQ and DPASQ. **a** The chemical structures of MeOSQ and DPASQ. **b** Photoluminescence (PL) spectra of MeOSQ in THF/water mixtures with different water fractions (f_w). Concentration (c) = 10^{-5} M, excitation wavelength (λ_{ex}) = 390 nm. **c** Plots of relative PL intensity (I/I_0) and maximum emission wavelength versus f_w . I_0 = PL intensity at $f_w = 0\%$. Inset: fluorescent photographs of MeOSQ in the THF/water mixtures with f_w of 0% and 90%, respectively. **d** PL spectra of DPASQ in THF/water mixtures with different f_w . $c = 10^{-5}$ M, $\lambda_{ex} = 430$ nm. **e** Plots of I/I_0 and maximum emission wavelength versus f_w . Inset: fluorescent photographs of DPASQ in the THF/water mixtures with water fractions of 60% and 90%, respectively.

Fig. 3 Photoluminescence (PL) properties of MQ, SQ, MeOSQ, and DPASQ in different solvents. **a-d** Normalized PL spectra of (a) MQ, (b) SQ, (c) MeOSQ, and (d) DPASQ in different solvents with increased polarity (*n*-hexane (hexane) < toluene < dioxane < tetrahydrofuran (THF) < dichloromethane (DCM) < dimethyl sulfoxide (DMSO) < acetonitrile (ACN) < methanol (MeOH)). Concentration = 10^{-5} M. The arrow indicates the increase of solvent polarity. **e-h** The relative PL intensity (I/I_{\min}) of (e) MQ, (f) SQ, (g) MeOSQ, and (h) DPASQ in different solvents versus the normalized Reichardt's parameter (E_T^N). I_{\min} = the lowest PL intensity of the target compound in different solvents. The linear fitting (red line) represents the trend of I/I_{\min} in different solvents with increased polarity, R^2 = goodness of fit.

REVIEWERS' COMMENTS

Reviewer #1 (Remarks to the Author):

The authors have included new experimental and computational results in answer to the main points raised during the first round of review. The inclusion of a methylated derivative nicely answers my questions about the possibility of tautomerism and dimerisation.

Based on these changes, as well as the additional data and corrections made in response to comments from other reviewers, I would now recommend this article for acceptance.

Reviewer #2 (Remarks to the Author):

The authors have revised the manuscript very carefully according to the suggestions. All the comments have been addressed. The revised version can be accepted.

Reviewer #3 (Remarks to the Author):

The authors submitted additional data and appropriately answered my queries. Therefore, I judged that this manuscript deserves publication as is.

Point-by-point Response to the Reviewers' Comments

Response to the Comments of Reviewer 1:

The authors have included new experimental and computational results in answer to the main points raised during the first round of review. The inclusion of a methylated derivative nicely answers my questions about the possibility of tautomerism and dimerisation.

Based on these changes, as well as the additional data and corrections made in response to comments from other reviewers, I would now recommend this article for acceptance.

Response: Thanks a lot for the reviewer's help and comments that help us improve the quality of the work.

Response to the Comments of Reviewer 2:

The authors have revised the manuscript very carefully according to the suggestions. All the comments have been addressed. The revised version can be accepted.

Response: Thanks a lot for the reviewer's kind help and valuable suggestions.

Response to the Comments of Reviewer 3:

The authors submitted additional data and appropriately answered my queries. Therefore, I judged that this manuscript deserves publication as is.

Response: Thanks a lot for the reviewer's recognition of our revisions and the revised manuscript.